# Chemoptogenetic ablation of neuronal mitochondria in vivo with spatiotemporal precision and controllable severity

Wenting Xie[1,2,3†], Binxuan Jiao[1,2,3†], Qing Bai[1,2†], Vladimir A Ilin[1,2†], Ming Sun[4], Charles E Burton[5], Dmytro Kolodieznyi[6], Michael J Calderon[4,7], Donna B Stolz[4,7], Patricia L Opresko[8,9], Claudette M St Croix[4,7], Simon Watkins[4,7], Bennett Van Houten[9,10], Marcel P Bruchez[6,11], Edward A Burton[1,2,12]*

[1]Department of Neurology, University of Pittsburgh, Pittsburgh, United States; [2]Pittsburgh Institute for Neurodegenerative Diseases, University of Pittsburgh, Pittsburgh, United States; [3]Tsinghua University Medical School, Beijing, China; [4]Center for Biologic Imaging, University of Pittsburgh, Pittsburgh, United States; [5]Winchester Thurston School, Pittsburgh, United States; [6]Departments of Biological Sciences and Chemistry, Carnegie Mellon University, Pittsburgh, United States; [7]Department of Cell Biology, University of Pittsburgh, Pittsburgh, United States; [8]Department of Environmental and Occupational Health, University of Pittsburgh, Pittsburgh, United States; [9]Genome Stability Program, UPMC Hillman Cancer Center, Pittsburgh, United States; [10]Department of Pharmacology and Chemical Biology, University of Pittsburgh, Pittsburgh, United States; [11]Molecular Biosensors and Imaging Center, Carnegie Mellon University, Pittsburgh, United States; [12]Geriatric Research, Education and Clinical Center, Pittsburgh VA Healthcare System, Pittsburgh, United States

*For correspondence:
eab25@pitt.edu

†These authors contributed
equally to this work

Competing interests: The
authors declare that no
competing interests exist.

Reviewing editor: Stephen C
Ekker, Mayo Clinic, United States

**Abstract** Mitochondrial dysfunction is implicated in the pathogenesis of multiple neurological diseases, but elucidation of underlying mechanisms is limited experimentally by the inability to damage specific mitochondria in defined neuronal groups. We developed a precision chemoptogenetic approach to target neuronal mitochondria in the intact nervous system in vivo. MG2I, a chemical fluorogen, produces singlet oxygen when bound to the fluorogen-activating protein dL5** and exposed to far-red light. Transgenic zebrafish expressing dL5** within neuronal mitochondria showed dramatic MG2I- and light-dependent neurobehavioral deficits, caused by neuronal bioenergetic crisis and acute neuronal depolarization. These abnormalities resulted from loss of neuronal respiration, associated with mitochondrial fragmentation, swelling and elimination of cristae. Remaining cellular ultrastructure was preserved initially, but cellular pathology downstream of mitochondrial damage eventually culminated in neuronal death. Our work provides powerful new chemoptogenetic tools for investigating mitochondrial homeostasis and pathophysiology and shows a direct relationship between mitochondrial function, neuronal biogenetics and whole-animal behavior.

## Introduction

High-resolution intravital imaging, coupled with transgenic expression of fluorescent reporters, provides opportunities to analyze the biology of the intact nervous system. This approach has been particularly successful in larval zebrafish, which combine optical transparency (*White et al., 2008*) with

**eLife digest** Most life processes require the energy produced by small cellular compartments called mitochondria. Many internal and external factors can harm these miniature powerhouses, potentially leading to cell death. For instance, in patients with Parkinson's or Alzheimer's disease, dying neurons often show mitochondrial damage. However, it is unclear exactly how injured mitochondria trigger the demise of these cells. Gaining a better understanding of this process requires studying the impact of mitochondrial damage in live neurons, something that is still difficult to do.

As a response to this challenge, Xie, Jiao, Bai, Ilin et al. designed a new tool that can specifically injure mitochondria in the neurons of live zebrafish larvae at will, and fine-tune the amount of damage inflicted. The zebrafish are genetically engineered so that the mitochondria in their neurons carry a protein which can bind to a chemical compound called MG2I. When attached to each other, MG2I and the protein respond to far-red light by locally creating highly damaging chemicals. This means that whenever far-red light is shone onto the larvae, mitochondria in their neurons are harmed – the brighter the light, the stronger the damage.

Zebrafish larvae exposed to these conditions immediately stopped swimming: mitochondria in their neurons could not produce enough energy and these cells could therefore no longer communicate properly. The neurons then started to die about 24 hours after exposure to the light, suggesting that the mitochondrial damage triggered other downstream processes that culminated in cell death.

This new light-controlled tool could help to understand the consequences of mitochondrial damage, potentially revealing new ways to rescue impaired neurons in patients with Parkinson's or Alzheimer's disease. In the future, the method could be adapted to work in any type of cell and deactivate other cell compartments, so that it can be used to study many types of diseases.

vertebrate CNS structure and genetics (*Burton, 2014*), allowing advances in understanding whole-brain activity patterns at cellular resolution (*Portugues et al., 2014*; *Ahrens et al., 2013*), the formation of neural circuits (*Tay et al., 2011*), neuronal mitochondrial trafficking (*Dukes et al., 2016*) and neuronal autophagy (*Khuansuwan et al., 2019*). Recent developments in light-sensitive channel proteins (*Cosentino et al., 2015*) and genetically-encoded photosensitizers (*Buckley et al., 2017*; *He et al., 2016*) provide new opportunities to extend these studies from observations to experimental manipulation of the CNS in vivo, by light-induced activation (*Ljunggren et al., 2014*), inhibition (*Bergeron et al., 2015*) or ablation (*Del Bene et al., 2010*) of genetically-defined neuronal populations. In particular, genetically-encoded photosensitizers should offer the means to analyze neuronal biology with stringent spatial resolution, by damaging or ablating defined neuronal populations, cellular subcompartments, organelles or even specific molecules precisely, to determine their roles in nervous system function and pathophysiology.

Genetically-encoded photosensitizers often show limited photostability, decreasing the amount of oxidative damage that can be induced by light exposure (*He et al., 2016*). In addition, they are typically excited by blue or green light, which shows limited penetration through living tissues and can cause direct, untargeted, phototoxicity (*He et al., 2016*). Furthermore, constitutively-active photosensitizers allow chronic phototoxicity from passive absorption of ambient light. Chemogenetic systems enable selective targeting of genetically-defined cells by expression of a designer receptor that is only activated after administration of a small molecule ligand that binds the receptor specifically (*Roth, 2016*). Far-red light-sensitive channel proteins enable activation in deep tissues – even through the intact skull (*Chuong et al., 2014*). For optogenetic applications in vivo, it would be highly advantageous to combine on-demand chemical activation of genetically-targeted cells with photoactivation using far-red light. dL5** is a modified single-chain antibody that can function as a fluorogen-activating protein (FAP) by binding malachite green (MG) fluorogens with picomolar affinity and activating their fluorescence thousands-fold (*Szent-Gyorgyi et al., 2008*; *Szent-Gyorgyi et al., 2013*). It was recently reported that a di-iodine substituted MG derivative, MG2I, becomes a potent photosensitizer on binding to dL5** (*He et al., 2016*). The dL5**-MG2I complex showed maximal excitation in the highly tissue-penetrant far-red light range, and produced singlet

oxygen ($^1O_2$, molecular dioxygen in its first electronically excited state) through inter-system crossing, with high quantum yield and with little photobleaching (*He et al., 2016*). Importantly, neither dL5** nor MG2I alone produced detectable $^1O_2$ under far-red illumination, and the dL5**-MG2I complex did not produce $^1O_2$ in the absence of light. This system offers several key advantages as a photosensitizer for use in neuroscience applications in vivo. $^1O_2$ is highly reactive with organic molecules and thus reacts within a short radius (<20 nm) of its site of formation in living systems (its lifetime is approximately 4µs in water [*Wilkinson et al., 1995*], but may be as low as 100ns in cells [*Moan and Berg, 1991*]). Consequently, oxidative damage can be provoked with a high degree of spatial precision by fusing dL5** to an appropriate protein or targeting sequence, to direct its expression to a specific sub-cellular region, organelle or protein complex (*He et al., 2016*). For example, dL5** fused to TRF1, a component of the shelterin complex, was employed recently to induce 8-oxoguanine formation specifically in the telomeric DNA of cultured cells (*Fouquerel et al., 2019*). Furthermore, as $^1O_2$ is generated only during far-red illumination, the onset of oxidative damage from dL5**-MG2I is temporally well-defined, and its severity and rate of induction can be regulated by adjusting light exposure time and power. Finally, dependence of chemoptogenetic production of $^1O_2$ on the presence of MG2I means that dL5**-expressing transgenic animals and cell lines can be generated, bred, handled and shipped in the absence of MG2I, without having to house them in darkness; this is a significant advantage over constitutively active genetic photosensitizers.

Alterations in neuronal mitochondrial function have been strongly linked to several neurodegenerative diseases (*Nguyen et al., 2019*; *Swerdlow, 2018*; *Carmo et al., 2018*) and there is significant interest in understanding how mitochondrial homeostasis and bioenergetics are maintained in neurons under physiological conditions and following mitochondrial damage. Prior work showed that dL5** could be expressed within the mitochondria of cultured HEK293 cells in vitro when fused to an appropriate targeting signal (*Qian et al., 2019*). Treatment of these cells with MG2I and exposure to far-red light resulted in decreased oxygen consumption, loss of respiratory chain activity, mitochondrial depolarization, compensatory glycolysis, and secondary ROS generation that was sufficient to cause oxidative telomere damage and cell cycle arrest (*Qian et al., 2019*). Unlike transformed cells in culture, however, neurons are generally dependent on mitochondrial function for ATP generation (reviewed in *Van Laar and Berman, 2013*) and this is known to influence the responses of neuronal mitochondria to perturbing stimuli (*Van Laar et al., 2011*). To investigate the consequences of mitochondrial damage in neurons in vivo, we expressed dL5** in the neuronal mitochondria of transgenic zebrafish. In the presence of MG2I, far red light provoked specific disruption of mitochondrial structure, respiration and ATP synthesis, resulting in dramatic neurophysiological and neurobehavioral abnormalities. This highly innovative approach has numerous applications as a tool for understanding neuronal bioenergetics and mitochondrial homeostasis, investigating mitochondrial mechanisms in disease pathogenesis, and manipulating neural circuitry to understand the biological basis of behavior.

## Results

### Construction of transgenic NeuMitoFAP zebrafish expressing dL5** in neuronal mitochondria

We generated transgenic zebrafish expressing the fluorogen-activating protein (FAP) dL5** (*Figure 1A*; *Szent-Gyorgyi et al., 2008*; *Szent-Gyorgyi et al., 2013*) fused to mitochondrial targeting sequences from human COX4 and COX8 (*Telmer et al., 2015*), and a fluorescent reporter, mCerulean3, to allow visualization of transgene expression in vivo (*Figure 1B*; dL5** and mCerulean3 are separated by a flexible GGGGSGGGGS linker to allow correct folding of each domain of the fusion protein). Transgenic lines were made using the bipartite Gal4/UAS system (*Asakawa et al., 2008*), so that mitochondrially-targeted dL5**-mCerulean3 could be expressed in any tissue or cell population of interest by crossing Tg(*UAS:COX4-COX8-dL5**-mCer3*) zebrafish with an appropriate tissue-specific Gal4 driver line, thereby enhancing the utility of this line for multiple future applications. For the present study, we generated a new Tg(*eno2:gal4FF*) line that expresses Gal4FF (an engineered Gal4-VP16 fusion protein with attenuated toxicity; *Asakawa et al., 2008*) widely in neurons, using a 12 kb *eno2* regulatory element that we reported previously (*Bai et al., 2007*). Double transgenic Tg(*eno2:gal4FF*)[pt425]; Tg(*UAS:COX4-COX8-dL5**-mCer3*)[pt427]

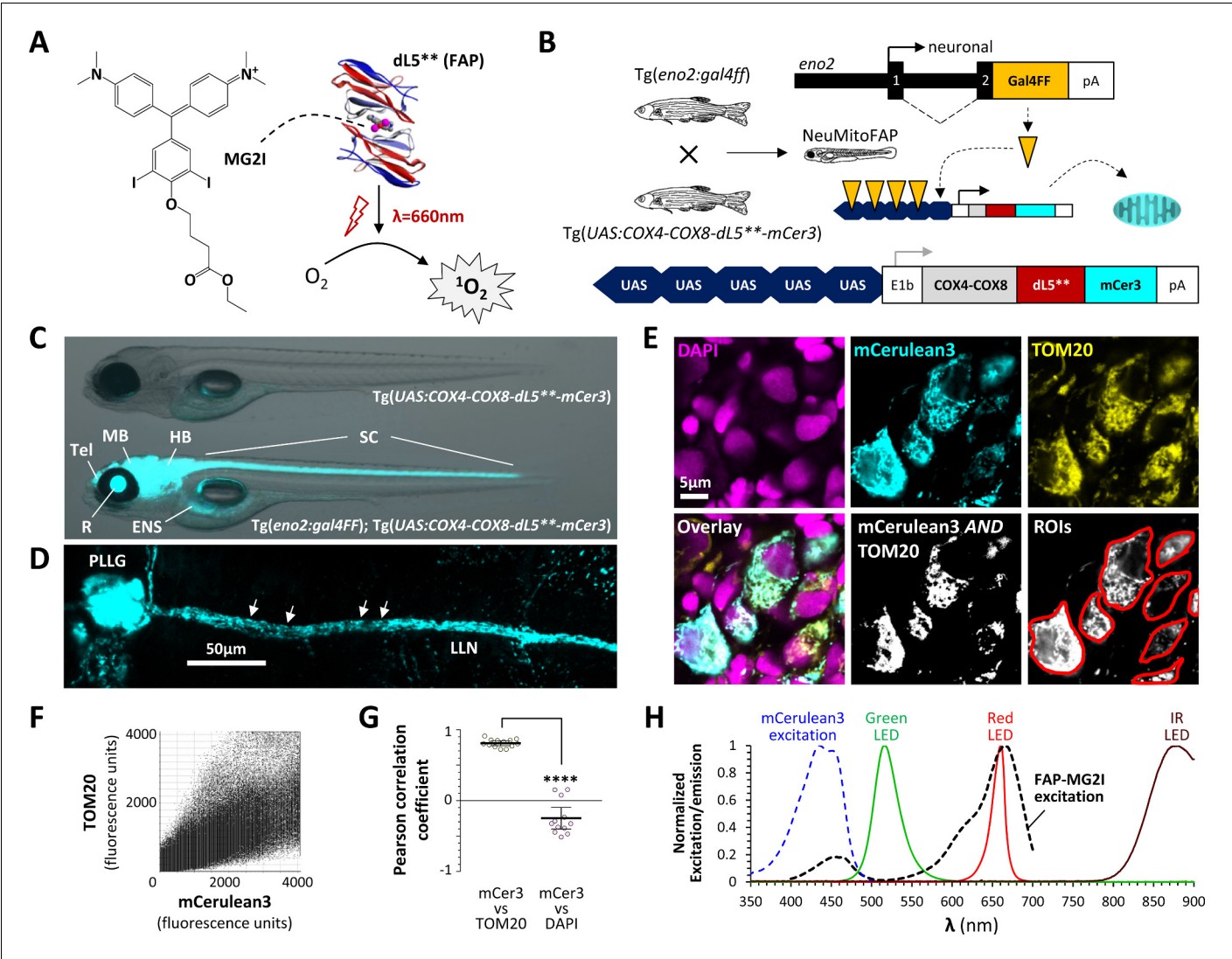

**Figure 1.** Generation of NeuMitoFAP zebrafish. (**A**) When the fluorogen MG2I (chemical structure shown on left) is bound to the fluorogen-activating protein (FAP) dL5** (right), excitation by far-red light causes generation of singlet oxygen. (**B**) Diagrams of transgene constructs eno2:gal4FF (above) and UAS:COX4-COX8-dL5**-mCer3 (below). Transactivation of the UAS enhancer by Gal4 in the neurons of double transgenic Tg(*eno2:gal4ff*); Tg(*UAS: COX4-COX8-dL5**-mCer3*) 'NeuMitoFAP' zebrafish results in expression of the dL5**-mCerulean3 fusion protein in the mitochondrial matrix. (**C**) Merged phase contrast and mCerulean3 epifluorescence images, showing live Tg(*UAS:COX4-COX8-dL5**-mCer3*) (above) and Tg(*eno2:gal4ff*); Tg(*UAS: COX4-COX8-dL5**-mCer3*) (NeuMitoFAP; below) zebrafish larvae at 5 days post-fertilization. mCerulean3-expressing structures are labeled (Tel, telencephalon; MB, midbrain; HB, hindbrain; SC, spinal cord; R, retina, ENS enteric nervous system). (**D**) Confocal z-plane projection showing mCerulean3 expression in the posterior lateral line ganglion (PLLG) and lateral line nerve (LLN) of a NeuMitoFAP zebrafish. Individual axonal mitochondria are indicated (arrows). (**E**) Brain sections from NeuMitoFAP zebrafish were labeled for nuclei (DAPI; magenta), dL5**-mCerulean3 (cyan) and mitochondria (TOM20; yellow). Single confocal planes of the individual channels are shown in the upper row. The lower row shows: the three channels overlaid; the output of a Boolean (mCerulean3 AND TOM20) map; and representative regions of interest that were analyzed in panels F and G. (**F**) Scatter plot of TOM20 signal (y-axis) versus mCerulean3 signal (x-axis) in each pixel within regions of interest. (**G**) Pearson correlation coefficient of signal intensity for mCerulean3 versus TOM20 (left) compared with mCerulean3 versus DAPI (right). Each data point shows a region of interest corresponding to an individual mCerulean3-expressing cell, bars show mean ± SE; ****p<0.0001, 2-tailed t-test. (**H**) Normalized excitation and emission spectra of the fluorophores and light sources used in this study.

The online version of this article includes the following figure supplement(s) for figure 1:

**Figure supplement 1.** High-power LED light source to deliver far-red light to larval zebrafish.

**Figure supplement 2.** Energy transfer spectra between LED sources and dL5**-MG2I.

**Figure supplement 3.** LED light source did not cause appreciable heating of water bath.

zebrafish (referred to as 'NeuMitoFAP' zebrafish for brevity) showed strong mCerulean3 expression throughout the nervous system (*Figure 1C*). At high magnification, punctate mCerulean3-labeled structures corresponding to individual mitochondria were visible within axons of live NeuMitoFAP larval neurons (*Figure 1D*). Tissue sections revealed extensive co-localization of mCerulean3 with TOM20 (a mitochondrial marker) in CNS neurons of NeuMitoFAP zebrafish (*Figure 1E*). Pixel-by-pixel analysis of single confocal planes showed that the mCerulean3 signal correlated strongly with the mitochondrial TOM20 signal, but not with the nuclear DAPI signal (*Figure 1F,G*). These data show that dL5**-mCerulean3 is expressed in the neuronal mitochondria of NeuMitoFAP zebrafish.

The excitation spectrum of the FAP-MG2I complex (*He et al., 2016*) is shown in *Figure 1H*, and summarized in *Table 1*, in comparison with emission spectra of the light sources employed in this study. A light stand was constructed (*Figure 1—figure supplement 1*) to expose zebrafish larvae to far-red light ($\lambda = 661 \pm 9$ nm, peak ±half width at half height; *Table 1*) near the major FAP-MG2I excitation peak ($\lambda = 666 \pm 30$ nm; *Figure 1—figure supplement 2*; *Table 2*), with adjustable power up to $\approx 160$ mW/cm$^2$, and without transferring heat to the water bath (*Figure 1—figure supplement 3*). Green LED safe lights ($\lambda = 516 \pm 18$ nm) allowed MG2I-exposed NeuMitoFAP zebrafish to be handled, and behavioral responses provoked (*Burton et al., 2017*), without activating $^1O_2$ production from the FAP-MG2I complex (*Figure 1H*; *Figure 1—figure supplement 2*; *Tables 1* and *2*). Infrared light sources that did not activate FAP-MG2I provided illumination for videography, while quantifying zebrafish motor function (*Zhou et al., 2014*) ($\lambda = 877 \pm 25$ nm) and during electrophysiological recordings ($\lambda = 775 \pm 32$ nm).

## Acute neurological deficits in NeuMitoFAP zebrafish exposed to MG2I and far-red light

By 5 days post-fertilization (dpf), zebrafish larvae show spontaneous locomotor activity and evoked behavioral responses that are easily quantified by video tracking in 96-well plates (*Figure 2*; *Zhou et al., 2014*). We previously demonstrated that the visual motor response (VMR), a series of stereotyped changes in motor activity provoked by abrupt alterations in ambient illumination (*Burgess and Granato, 2007*), can be elicited by green light at a wavelength that does not excite FAP-MG2I (*Burton et al., 2017*). Four experimental groups (WT, WT-MG2I, NeuMitoFAP, NeuMito-FAP-MG2I) were generated by growing non-transgenic and NeuMitoFAP zebrafish in embryo water containing MG2I or no additive under green light illumination from 3 dpf (*Figure 2*). Zebrafish from all four experimental groups showed normal motor activity (*Figure 2B*) and morphology (*Figure 2—figure supplement 1*) at 5dpf. Robust responses to abrupt green light – dark transitions were apparent in all four experimental groups (*Figure 2B*, left panel; *Figure 2C*, upper graphs) prior to far-red light exposure. However, motor responses were eliminated acutely in NeuMitoFAP-MG2I zebrafish, but not controls, following exposure to 60 J/cm$^2$ far-red light (mean ± SE swimming speed in dark phase of VMR, pre- versus post-exposure: WT, 1.80 ± 0.10 vs. 1.70 ± 0.08 mm/s; WT-MG2I, 1.69 ± 0.09 vs. 1.60 ± 0.10 mm/s; NeuMitoFAP, 2.02 ± 0.10 vs. 2.13 ± 0.08 mm/s; NeuMitoFAP-MG2I 2.36 ± 0.09 vs. 0.08 ± 0.02 mm/s, $p < 10^{-15}$, 2-way ANOVA with Šidák multiple comparisons test; *Figure 2B-D*; *Video 1*). Loss of motor function in NeuMitoFAP-MG2I zebrafish was dependent on the amount of light energy delivered (*Figure 2—figure supplement 1*), but independent of the rate of delivery between 16–160 mW/cm$^2$ at a fixed total exposure of 60 J/cm$^2$ (*Figure 2—figure supplement 2*). Motor function during FAP-MG2I activation was examined by using far-red light to both elicit the VMR and excite FAP-MG2I simultaneously (*Figure 2—figure supplement 3*). In Neu-MitoFAP-MG2I zebrafish, far-red light initially caused transient hyperkinesia, which was followed by

**Table 1.** Peak wavelength, centroid and full width at half height (FWHH) are shown for the red, green, and infrared LED sources used in the study, in comparison with the major and minor excitation peaks of the dL5**-MG2I complex (see *Figure 1H*).

|  | Red | Green | IR | dL5**-MG2I (minor) | dL5**-MG2I (major) |
|---|---|---|---|---|---|
| Peak λ (nm) | 661 | 516 | 877 | 456 | 666 |
| Centroid λ (nm) | 656 | 520 | 880 | 452 | 649 |
| FWHH (nm) | 18 | 36 | 51 | 55 | 61 |

**Table 2.** Peak wavelength, peak height and area under the curve are shown for normalized energy transfer spectra between the 661 nm LED and the dL5**-MG2I complex ('Red x dL5**-MG2I') and between the 516 nm LED and the dL5**-MG2I complex ('Green x dL5**-MG2I'; see *Figure 1— figure supplement 2*).

|  | Red x dL5**-MG2I | Green x dL5**-MG2I |
|---|---|---|
| Peak λ (nm) | 661 | 524 |
| Height | 0.98 | 0.013 |
| Area | 19.10 | 1.15 |

progressive loss of motor function with cumulative light exposure; these abnormalities were not observed in controls. Although NeuMitoFAP-MG2I zebrafish exposed to far-red light lost spontaneous and evoked motor function, their heart rate and circulation were preserved (*Figure 2E*; *Video 2*) and there were no gross morphological changes (*Figure 2—figure supplement 4*), showing that the larvae remained alive and that abnormalities were restricted to the nervous system. Together, these data show a dramatic neurological phenotype that was dependent on all three components of the chemoptogenetic system (far-red light exposure in the presence of both dL5** and MG2I), and therefore attributable to $^1O_2$ formation at the site of dL5** expression in neuronal mitochondria. Furthermore, the severity of the abnormalities was determined by the amount of far-red light energy delivered.

### Acute neuronal depolarization caused by rapid loss of ATP in NeuMitoFAP zebrafish exposed to MG2I and far-red light

Electrophysiological recordings were carried out to elucidate the basis for the abrupt loss of neurological function in NeuMitoFAP-MG2I zebrafish exposed to far-red light (*Figure 3*; *Figure 3—figure supplement 1*). Large sensory neurons of the posterior lateral line ganglion express the *eno2* driver strongly and are located superficially, allowing continuous whole-cell patch clamp recordings in intact zebrafish larvae during light exposure. In control zebrafish, lateral line sensory neurons showed similar stable baseline membrane potentials that changed minimally during 20–30 min of far-red light exposure (baseline versus final membrane potential: WT-MG2I, −66.9 ± 1.4 vs. −63.2 ± 1.2 mV; NeuMitoFAP, −64.4 ± 1.2 vs. −60.8 ± 2.5 mV; mean ± SE; *Figure 3A-C*). Sensory neurons in NeuMitoFAP-MG2I larvae showed similar baseline membrane potential to controls but depolarized progressively during exposure to far-red light, initially reaching threshold potential and firing high-frequency trains of action potentials, then depolarizing further to become refractory (baseline versus final membrane potential −60.8 ± 2.4 mV vs. −29.4 ± 4.5 mV, p<0.0001, 2-way ANOVA with Tukey multiple comparison test; *Figure 3A-C*). To determine whether bioenergetic depletion could account for these findings, identical recordings were obtained from NeuMitoFAP-MG2I neurons, but with the addition of phosphocreatine to the patch pipette solution. Phosphocreatine is a substrate for cytoplasmic creatine kinase, allowing the regeneration of ATP from ADP and thereby providing a non-mitochondrial source of ATP. The baseline resting membrane potential was unaltered by phosphocreatine, but its presence prevented depolarization of NeuMitoFAP-MG2I neurons during far-red light exposure (baseline versus final membrane potential: −62.7 ± 1.4 vs. −59.6 ± 2.8 mV; *Figure 3A-C*). Overall, 10/10 NeuMitoFAP-MG2I neurons depolarized by ≥20% of their baseline membrane potential following a mean exposure of 44.3 ± 6.3 J/cm$^2$ far-red light, whereas in the presence of phosphocreatine, only 1/6 neurons depolarized after exposure to >80 J/cm$^2$ (p=0.0014, Fisher's exact test). Transient exposures to smaller amounts of far-red light energy caused slower and more modest levels of depolarization (*Figure 3—figure supplement 2*). Together, these data show that the acute neurological deficits provoked in NeuMitoFAP-MG2I zebrafish by exposure to far-red light were caused by neuronal depolarization, resulting from depletion of ATP that is necessary to drive the active ionic transporters that maintain transmembrane ionic gradients underlying the resting membrane potential.

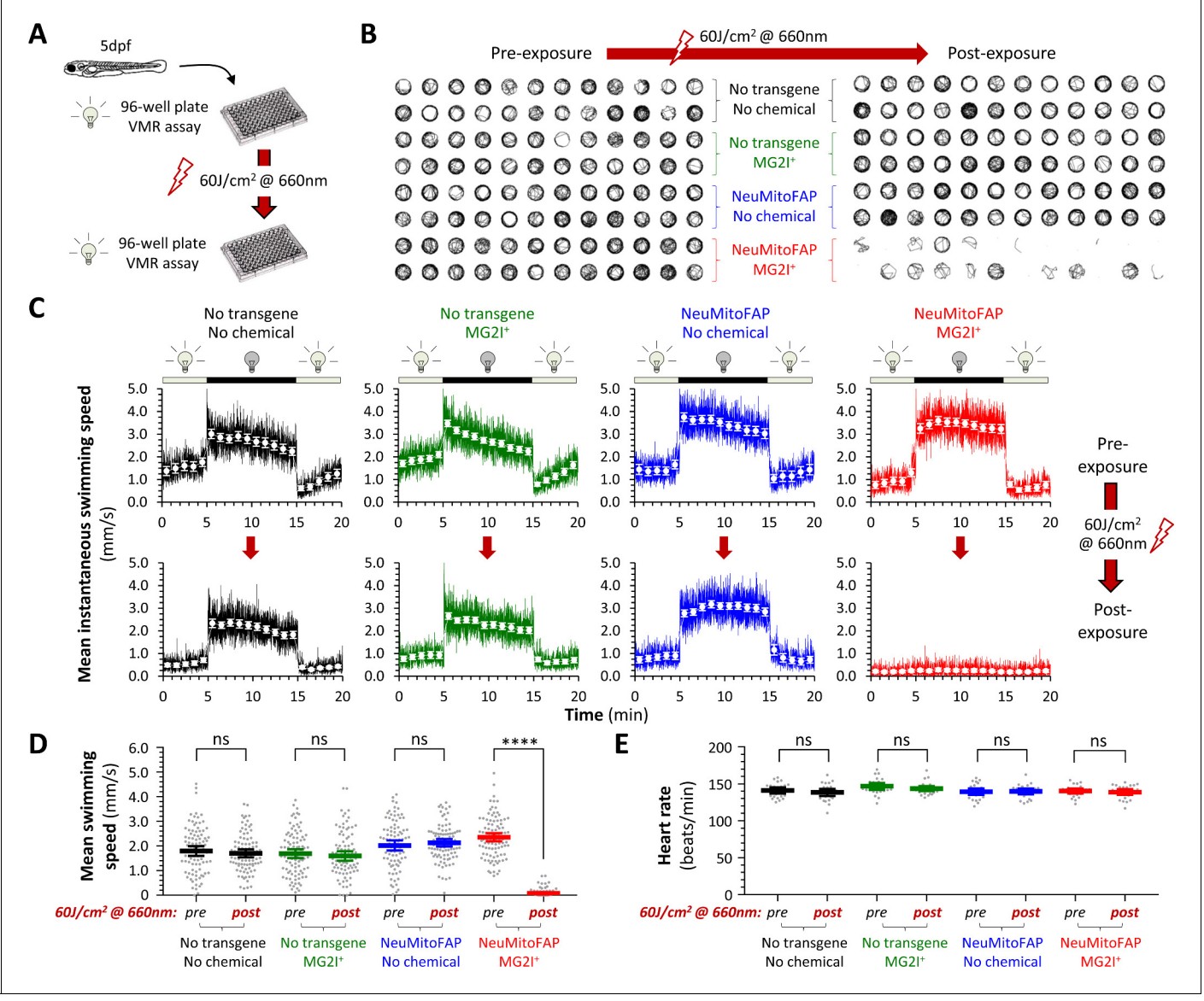

**Figure 2.** Acute loss of neurological function in NeuMitoFAP zebrafish exposed to MG2I and far-red light. (**A**) Design of experiments shown in panels (**B** - **D**). Experimental groups: WT zebrafish (black); WT zebrafish exposed to MG2I (green); NeuMitoFAP zebrafish (blue); NeuMitoFAP zebrafish exposed to MG2I (red). (**B**) 1 min swimming vectors of zebrafish in a 96-well plate, before (left) and after (right) exposure to 60 J/cm² far-red light (160 mW, $\lambda_{peak}$=661nm). Each group includes 24 zebrafish, occupying two rows of the 96-well plate as indicated. (**C**) The visual motor response (VMR) was elicited by alternating 10 min periods of green light illumination (250 Lux, $\lambda_{peak}$=516 nm; does not activate dL5**-MG2I) and darkness (0 Lux). The graphs show mean instantaneous group swimming speed (y-axis) against time (x-axis). Responses were averaged over three cycles of dark/light stimuli (solid colored lines; illumination cycle shown above each graph); responses of individual zebrafish were further averaged into 1 min time bins (white markers, error bars show ±95% CI). Responses of the same zebrafish are shown before (top row) and after (bottom row) exposure to 60 J/cm² far-red light. n = 24 zebrafish/group. (**D**) Mean swimming speed during the dark phase of the VMR (y-axis) was quantified before and after exposure to 60 J/cm² far-red light. Data points show responses of individual zebrafish (n = 96/group from four combined replicate experiments, bars show mean ± 95% CI). ****$p < 10^{-15}$, pre- versus post-light exposure swimming speed, 2-way ANOVA with Šidák multiple comparisons test. (**E**) Heart rate (y-axis) was quantified before and after exposure to 60 J/cm² far-red light (data points show individual zebrafish, bars show mean ± 95% CI).

The online version of this article includes the following source data and figure supplement(s) for figure 2:

**Source data 1.** Source data for *Figure 2C*.
**Source data 2.** Source data for *Figure 2E*.
**Figure supplement 1.** Normal morphology of NeuMitoFAP zebrafish exposed to MG2I and light.
**Figure supplement 2.** The motor phenotype of NeuMitoFAP-MG2I zebrafish is dependent on far-red light exposure energy.
**Figure supplement 3.** The motor phenotype of NeuMitoFAP-MG2I zebrafish exposed to 60 J/cm² far-red light is not dependent on light power.
**Figure supplement 4.** Real-time changes in motor function of NeuMitoFAP-MG2I zebrafish during far-red light exposure.

## Loss of neuronal mitochondrial function in NeuMitoFAP zebrafish exposed to MG2I and far-red light

We next evaluated mitochondrial function in NeuMitoFAP-MG2I zebrafish. A bioluminescence assay was used to measure ATP concentration in lysates from NeuMitoFAP-MG2I larvae. Far-red light exposure caused a ≈10% decrease in whole animal ATP content (pre-light 4.12 ± 0.10 vs. post-light 3.69 ± 0.11 µMol ATP/g protein; mean ± SE; p=0.0060, 2-tailed t-test; *Figure 4A*). Mitochondrial respiration was analyzed using a flux analyzer to measure whole larval $O_2$ consumption rate (OCR). In order to allow stable recordings, and to prevent changes in muscle $O_2$ consumption secondary to altered motor activity from obscuring differences between experimental groups, larvae were anesthetized with tricaine and paralyzed with d-tubocurarine prior to OCR quantification (*Figure 4B*; *Figure 4—figure supplement 1*). NeuMitoFAP zebrafish exposed to far-red light in the presence of MG2I showed a ≈24% decrease in whole larval OCR compared with controls that were not treated with MG2I (NeuMitoFAP 170.7 ± 8.9 vs. NeuMitoFAP-MG2I 129.2 ± 7.0 pMol/min; mean ±SE; p=0.00053, unpaired t-test; *Figure 4C,D*). In contrast, WT controls exposed to far-red light showed similar stable OCRs regardless of MG2I treatment (WT, 194.8 ± 11.4; WT-MG2I, 189.1 ± 14.1 pMol/min; mean ± SE; *Figure 4E*). Bath acidification rate was unaltered in NeuMitoFAP-MG2I zebrafish exposed to far-red light (*Figure 4—figure supplement 2*), providing no evidence of compensatory glycolysis following abrogation of neuronal mitochondrial function. Cultured cells, unlike whole zebrafish, can be treated with chemical inhibitors to evaluate specific aspects of mitochondrial function. Consequently, we next evaluated dissociated brain cells derived from NeuMitoFAP zebrafish. Cells were treated with MG2I and exposed to far-red light after dissociation (*Figure 4F*). Under these conditions, NeuMitoFAP-MG2I cells, but not controls, showed a dramatic, far-red light

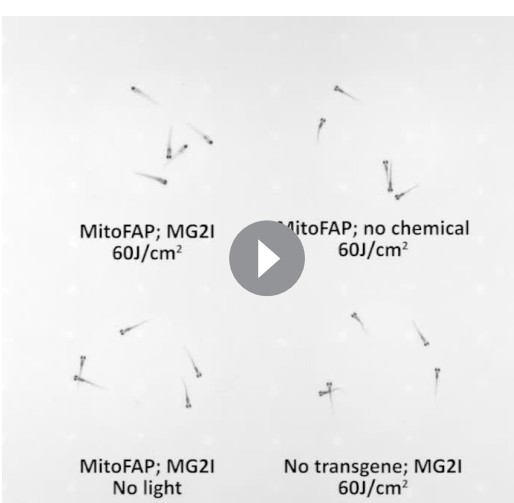

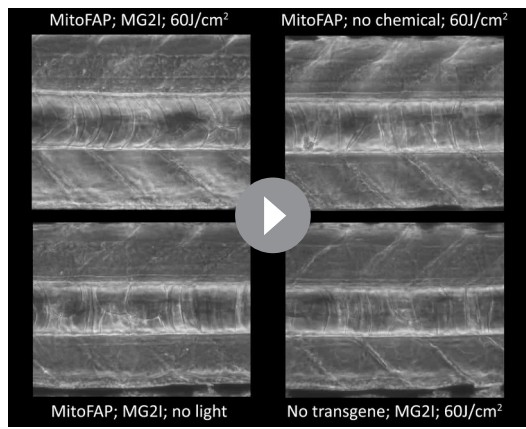

**Video 1.** Loss of motor function in NeuMitoFAP-MG2I zebrafish following far-red light exposure. 5dpf zebrafish swimming in the wells of an agarose-filled plate were illuminated from above by an infrared light, and video was recorded from below at 30 frame/s, during the dark phase of the visual motor response (VMR). Each well contains 5 zebrafish from the experimental groups indicated: *top left* – NeuMitoFAP zebrafish exposed to MG2I and 60 J/cm² far-red light; *top right* – NeuMitoFAP zebrafish exposed to 60 J/cm² far-red light in the absence of MG2I; *bottom left* – NeuMitoFAP zebrafish treated with MG2I but not exposed to far-red light; bottom right – WT zebrafish exposed to MG2I and 60 J/cm² far-red light. Dramatic abnormalities of motor function are clearly visible in the NeuMitoFAP-MG2I-light group, but not in any of the controls.

https://elifesciences.org/articles/51845#video1

**Video 2.** Normal circulation and heartbeat in NeuMitoFAP-MG2I zebrafish following far-red light exposure. (i) Phase contrast videomicrography of 5dpf zebrafish from the same experimental groups as *Video 1*, illustrating identical circulation of blood cells in the vascular system from all experimental groups (zebrafish oriented rostral towards left); (ii) high-power videomicrography from a zebrafish in the NeuMitoFAP-MG2I-light group that showed profound loss of neurological function. Anatomical landmarks are labeled for orientation (rostral left). (iii) The heart is shown from zebrafish in all four experimental groups, illustrating normal contraction and heart rate. These data show that abnormalities in NeuMitoFAP zebrafish exposed to MG2I and far-red light are confined to the nervous system and that the zebrafish remain alive and viable despite profound neurological disruption.

https://elifesciences.org/articles/51845#video2

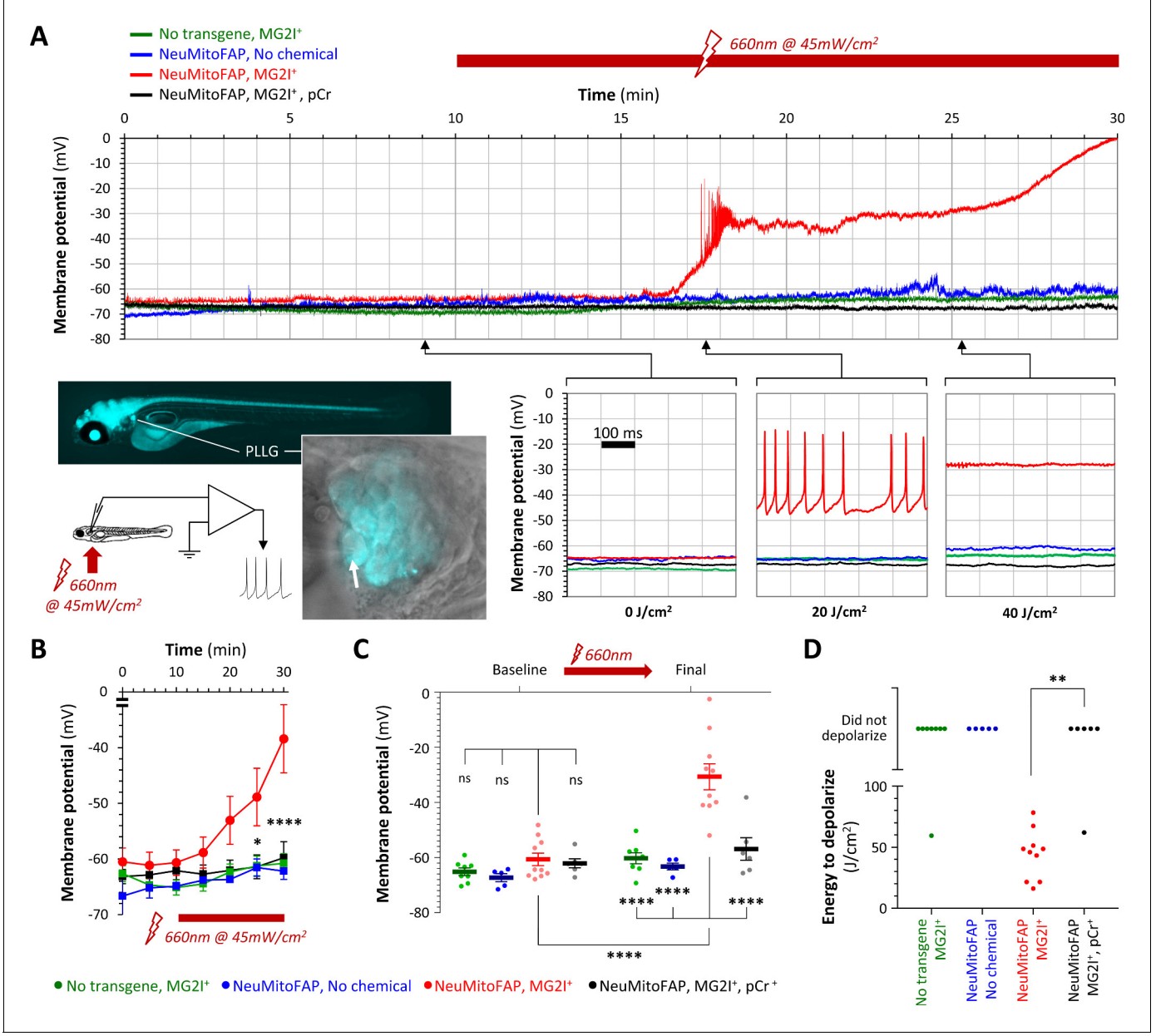

**Figure 3.** Acute neuronal depolarization in NeuMitoFAP zebrafish exposed to MG2I and far-red light. (A) Whole-cell patch clamp recordings were made from posterior lateral line ganglion (PLLG) sensory neurons. The inset figure shows the preparation and experimental design (the arrow shows a patch-clamp pipette in contact with a mitoFAP-expressing neuron). Recordings were made for 30–40 min (10 min in darkness and then a further 20–30 min under illumination in far-red light at $\lambda_{peak}$=661 nm). The graphs show example traces of membrane potential (y-axis) against time (x-axis), for the full 30 min recording (upper graph) and for three 500 ms sweeps after cumulative far-red light doses of 0, 20 or 40 J/cm² as indicated. Experimental groups: WT zebrafish exposed to MG2I (green); NeuMitoFAP zebrafish (blue); NeuMitoFAP zebrafish exposed to MG2I (red); NeuMitoFAP zebrafish exposed to MG2I, recordings made with phosphocreatine (pCr) added to pipette solution (black). (B) Mean ± SE membrane potential (y-axis; 5–10 neurons per group) in each 5 min time bin (x-axis) during recording. *p<0.05, ****p<0.0001, NeuMitoFAP-MG2I versus other groups at same time point, 2-way repeated measures ANOVA with Tukey multiple comparisons test. (C) Membrane potential of lateral line ganglion neurons (y-axis) at baseline and final potential after far-red light exposure. Data points show individual neurons, bars show mean ± SE. ****p<0.0001, 1-way ANOVA with Tukey multiple comparison test. (D) Amount of far-red light energy necessary to decrease membrane potential by >20% from baseline value (y-axis). Data points show individual neurons. Neurons that did not depolarize during the recording period are shown above the graph. **p<0.01, Fisher's exact test. The online version of this article includes the following source data and figure supplement(s) for figure 3:

**Source data 1.** Source data for *Figure 3B-D*.
**Figure supplement 1.** Far-red light-induced depolarization of lateral line ganglion neurons in NeuMitoFAP zebrafish treated with MG2I.
**Figure supplement 2.** Severity of lateral line ganglion neuron depolarization in NeuMitoFAP-MG2I zebrafish is dependent on far-red light dose.

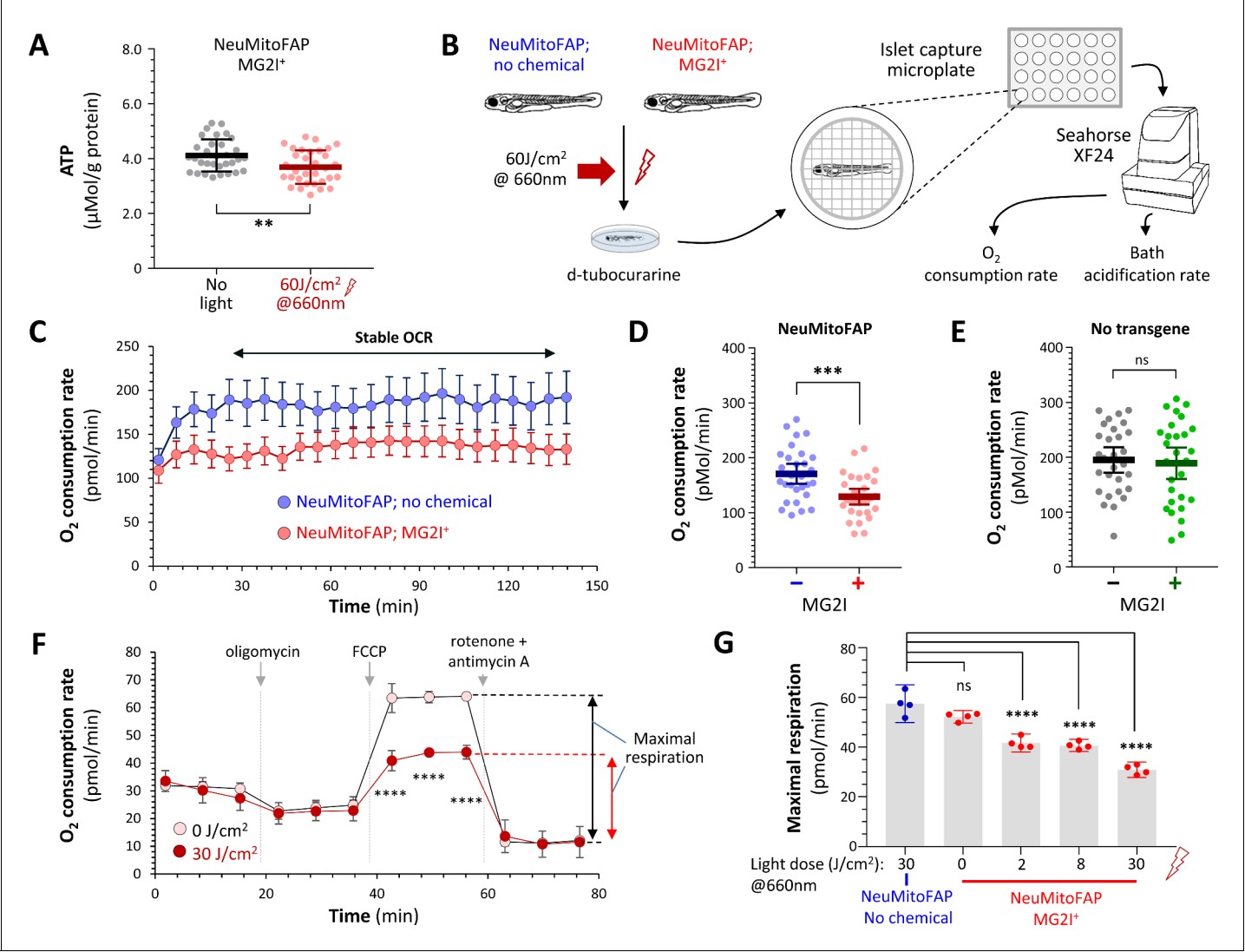

**Figure 4.** Disruption of mitochondrial function in NeuMitoFAP zebrafish exposed to MG2I and far-red light. (**A**) ATP concentration was measured in lysates (each containing 5 whole zebrafish larvae) using a bioluminescent assay and normalized to protein content. Data points show ATP/protein in individual lysates, bars show group mean ± 95% CI. Experimental groups: NeuMitoFAP zebrafish treated with MG2I (black); NeuMitoFAP zebrafish treated with MG2I and exposed to 60 J/cm² far-red light (red). **p<0.006, unpaired 2-tailed t-test. Graphs show results from four combined replicate experiments (n = 32 per group). (**B**) Design of experiments shown in panels C – E and *Figure 4—figure supplement 2*. (**C**) Oxygen consumption rate (OCR; y-axis) against time (x-axis) for NeuMitoFAP zebrafish exposed to far-red light in the presence (red) or absence (blue) of MG2I. Points show mean ± SE (n = 8 zebrafish larvae per group) in a single experiment. The period of stable OCR analyzed in panels D and E is indicated. (**D, E**) Stable OCR for (**D**) NeuMitoFAP and (**E**) WT zebrafish exposed to far-red light in the presence (+) or absence (−) of MG2I. Data points show values for individual zebrafish larvae. Bars show mean ± SE for 24 zebrafish per group, combined from 3 replicate experiments. ***p=0.00053, 2-tailed unpaired t-test. (**F**) OCR measurements for dissociated brain cells derived from NeuMitoFAP zebrafish. Cells were exposed to MG2I and far-red light (filled circles) or no light (open circles) after dissociation; OCR was measured dynamically at baseline and after exposure to oligomycin (ATP synthase inhibitor; shows proportion of OCR linked to ATP synthesis), FCCP (mitochondrial uncoupler; allows calculation of maximal respiration) and rotenone + antimycin-A (complex I and III inhibitors; shows proportion of measured OCR attributable to mitochondrial respiration). The calculation for maximal respiratory rate = $OCR_{(FCCP)} - OCR_{(rotenone\ +\ antimycin-A)}$ is shown schematically to the right of the graph. Data points show mean ± SE for five samples per group. ***p<0.001, no light versus 30 J/cm², 2-way repeated measures ANOVA with Šidák multiple comparisons test. (**G**) Dose-response curve for maximal respiration versus far-red light dose in NeuMitoFAP dissociated brain cells in the presence (right, red) or absence (left, blue) of MG2I. Bars show mean ± 95% CI, data points show replicate assays n = 5 per point. ****p<0.0001 versus no chemical control, 1-way ANOVA with Dunnett's *post hoc* test.

The online version of this article includes the following source data and figure supplement(s) for figure 4:

**Source data 1.** Source data for *Figure 4A*.
**Source data 2.** Source data for *Figure 4B*.
*Figure 4 continued on next page*

*Figure 4 continued*

**Source data 3.** Source data for *Figure 4D-E*.
**Source data 4.** Source data for *Figure 4F*.
**Figure supplement 1.** Viability of zebrafish larvae following neuromuscular paralysis with curare.
**Figure supplement 2.** No compensatory increase in glycolysis after mitochondrial targeting in NeuMitoFAP zebrafish.

dose-dependent, decrease in maximal respiration (measured by comparing OCR following exposure to the uncoupling agent FCCP with OCR following exposure to the complex I and III inhibitors rotenone and antimycin-A; *Figure 4F,G*). Together, these data show that the combination of MG2I and far-red light caused significant disruption of mitochondrial bioenergetic and respiratory functions in NeuMitoFAP zebrafish.

## Disruption of neuronal mitochondrial structure in NeuMitoFAP zebrafish exposed to MG2I and far-red light

We next investigated mitochondrial morphology by intravital microscopy. The lateral line nerve runs superficially along the larval body axis in a rostro-caudal direction, allowing confocal imaging at sufficiently high magnification to visualize the morphology of mCerulean3-labeled mitochondria within the axons of live, intact NeuMitoFAP zebrafish (*Figure 5A*). Image stacks were acquired through the entire medio-lateral extent of the nerve and mitochondrial features analyzed quantitatively. At baseline, mitochondria were elongated in shape and distributed regularly along axons (length $4.89 \pm 0.34$ µm; circularity $0.29 \pm 0.017$; number of mitochondria per field of view $75.50 \pm 6.84$; mean $\pm$ SE; *Figure 5A-D*). Length, shape and distribution were unaffected by either far-red light or MG2I alone (*Figure 5A–D*). However, in the presence of MG2I, an increased number of small, rounded mitochondria were seen in NeuMitoFAP axons immediately following far-red light exposure (length $2.00 \pm 0.12$ µm, p<0.0001; circularity $0.60 \pm 0.014$, p<0.0001; number $131.70 \pm 10.57$, p<0.001; NeuMitoFAP-MG2I post-light compared indivdually with each control group, 1-way ANOVA with Tukey multiple comparisons test; *Figure 5A-D*). These data suggest that the combination of NeuMitoFAP, MG2I and far-red light caused fragmentation of axonal mitochondria through mitochondrial $^1O_2$ production.

Transmission electron microscopy was employed to investigate mitochondrial ultrastructure in the brains of NeuMitoFAP zebrafish, immediately following far-red light exposure. In the absence of MG2I, neurons showed normal morphology; their mitochondria were elongated in shape and filled with densely-stacked tubular and lamellar cristae (*Figure 5E*). In the presence of MG2I, far-red light exposure caused the widespread appearance of lucent areas in neuronal cytoplasm at low magnification (*Figure 5F*, left panel, arrows). These areas corresponded to swollen, rounded mitochondria with nearly complete elimination of cristae (*Figure 5F*; *Figure 5—figure supplement 1*). The proportion of abnormal mitochondria showing rounded shape and absent or reduced cristae was determined in 12 fields of view per experimental group by a blinded observer ($6.14 \pm 2.84$ mitochondria per field, mean $\pm$ SD). No abnormal mitochondria were observed in NeuMitoFAP zebrafish prior to far-red light exposure. In the absence of MG2I, a single abnormal mitochondrion was seen immediately following far-red light exposure. In contrast, in the presence of MG2I, far-red light caused extensive disruption to neuronal mitochondrial ultrastructure. NeuMitoFAP-MG2I zebrafish showed $93.3 \pm 3.5\%$ abnormal mitochondria immediately post-exposure, and $97.0 \pm 3.0\%$ abnormal mitochondria at 24 hr post-exposure (p<$10^{-15}$ compared with no chemical control group at same time point; 2-way ANOVA with Šidák multiple comparisons test; *Figure 5G*). Together, these data show that far-red light exposure in the presence of MG2I caused severe structural deficits in the neuronal mitochondria of NeuMitoFAP zebrafish that were attributable to damage caused by mitochondrial chemoptogenetic $^1O_2$ production.

## Mitochondrial damage causes delayed neuronal death

Finally, we determined the downstream consequences of neuronal mitochondrial damage. The immediate loss of motor function observed in NeuMitoFAP-MG2I larvae following far-red light exposure did not recover over the subsequent 4 days, even though the zebrafish appeared morphologically unremarkable (*Figure 2—figure supplement 4*), underwent normal somatic, cardiac and

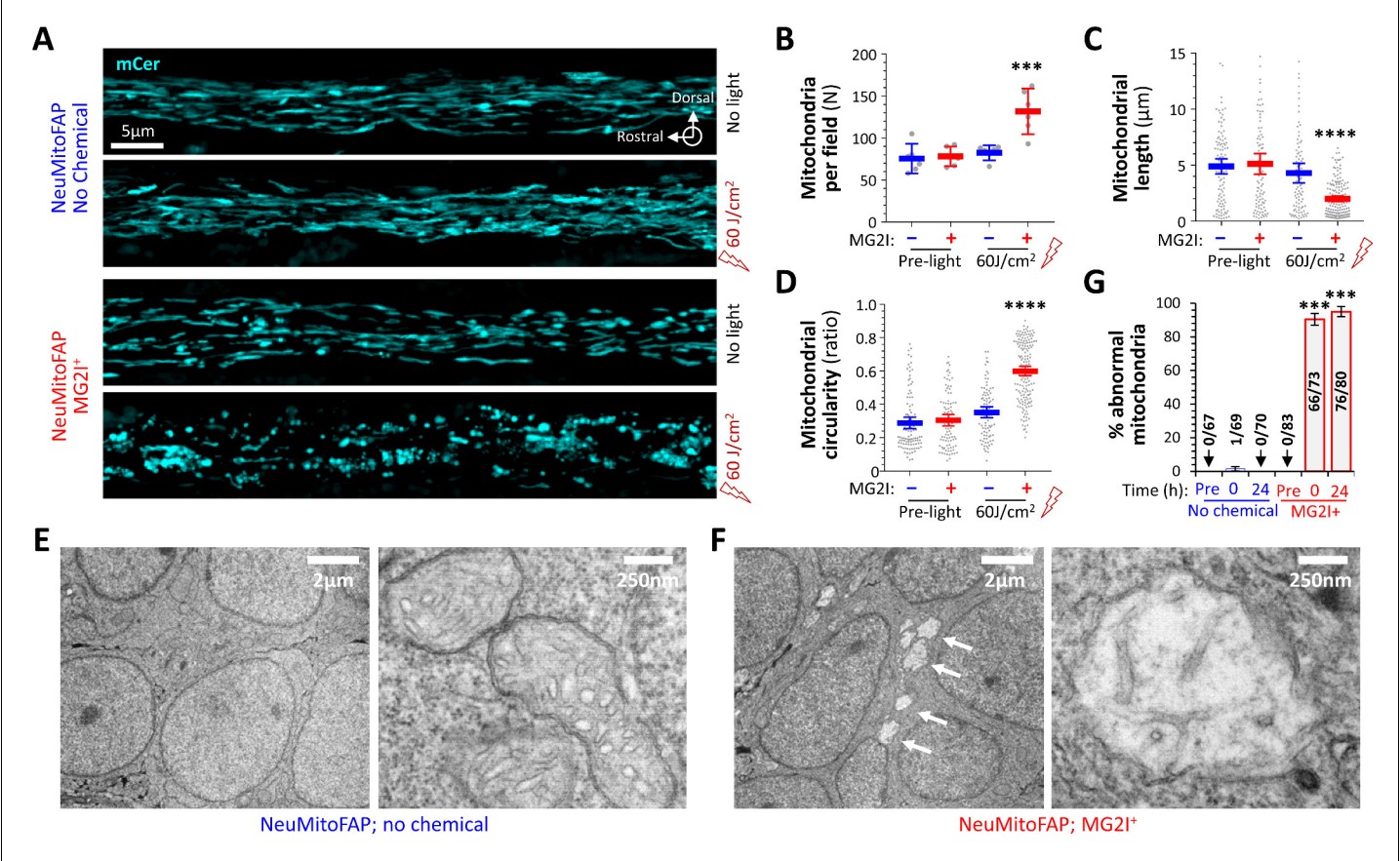

**Figure 5.** Disruption of mitochondrial structure in NeuMitoFAP zebrafish exposed to MG2I and far-red light. (**A**) Confocal Z-plane projections showing mCerulean-labeled mitochondria in the lateral line nerves of live NeuMitoFAP zebrafish in the absence (upper images) or presence (lower images) of MG2I, before (upper image of each pair) or after (lower image of each pair) exposure to 60 J/cm$^2$ light at $\lambda_{peak}$=661 nm. (**B - D**) The (**B**) number of mitochondria per field, (**C**) mitochondrial length, and (**D**) mitochondrial circularity were quantified in z-plane projections of the entire medio-lateral extent of the lateral line nerve in 6 zebrafish per group. Bars show mean ± 95% CI for NeuMitoFAP zebrafish in the absence (blue) or presence (red) of MG2I, before or after exposure to far-red light. Data points show individual zebrafish (panel B) or individual mitochondria (panels C, D). ***p<0.001, ****p<0.0001, NeuMitoFAP-MG2I post-light versus each other group individually, 1-way ANOVA with Tukey multiple comparisons test. (**E, F**) Transmission electron micrographs of sections from the telencephalon of NeuMitoFAP zebrafish immediately after exposure to far-red light in the (**F**) presence or (**E**) absence of MG2I. The left image of each pair shows a low-magnification view, and the right image shows a high-magnification view illustrating the ultrastructure of individual neuronal mitochondria. Arrows in panel F show swollen, damaged mitochondria. (**G**) The proportion of abnormal mitochondria was quantified by a blinded observer in 12 electron micrographs each from 6 experimental groups (NeuMitoFAP with or without MG2I, before, 0 or 24 hr after 60 J/cm$^2$ far-red light exposure). Bars show mean ± SE% abnormal mitochondria per section in each group. Numbers of total and abnormal mitochondria in each group are shown. ***p<0.001, compared individually with each control (pre-light and no chemical) group, 1-way ANOVA with Tukey multiple comparisons test.

The online version of this article includes the following source data and figure supplement(s) for figure 5:

**Source data 1.** Source data for *Figure 5B*.
**Source data 2.** Source data for *Figure 5C*.
**Source data 3.** Source data for *Figure 5D*.
**Source data 4.** Source data for *Figure 5G*.
**Figure supplement 1.** Widespread disruption of mitochondrial morphology in NeuMitoFAP zebrafish exposed to MG2I and far-red light.

vascular development and showed similar heartbeat and circulation to controls (*Figure 6A, B*; *Videos 3* and *4*). In order to clarify the basis for this persistent neurological deficit, we employed acridine orange (AO, a DNA intercalating agent that labels degenerating cells in which the plasma membrane has become permeable), in combination with intravital microscopy, to quantify cell death in live, intact larvae (*Figure 6C, D*; *Figure 6—figure supplement 1*). Baseline developmental cell death was observed in the spinal cords of all zebrafish (WT-MG2I, 9.5 ± 0.7; NeuMitoFAP,

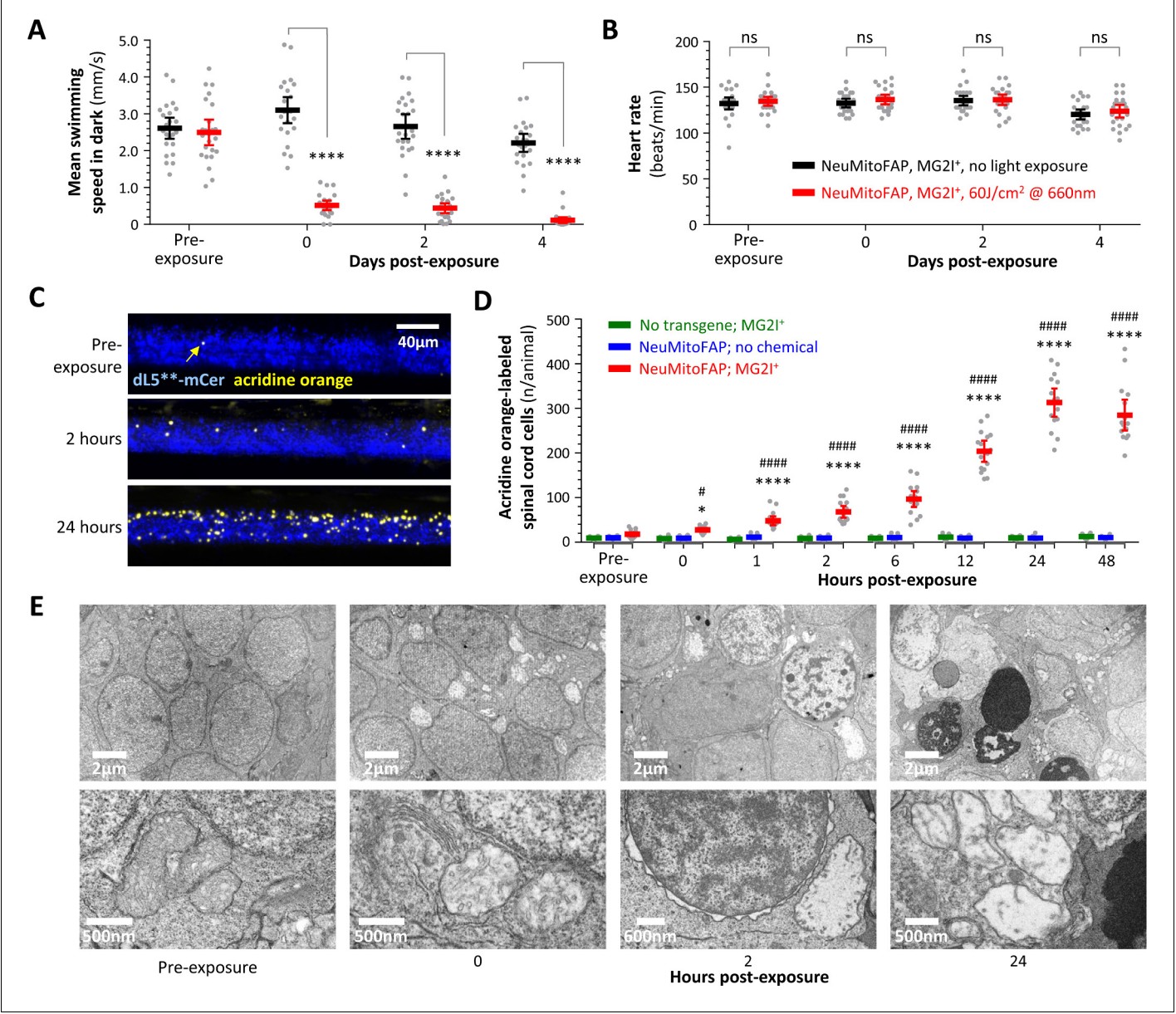

**Figure 6.** Persistent neurological deficits and cell death in NeuMitoFAP zebrafish exposed to MG2I and far-red light. (**A**) Mean swimming speed of NeuMitoFAP zebrafish treated with MG2I was quantified during the dark phase of the visual motor response (y-axis) after exposure to 60 J/cm² far-red light (red) or no exposure (black). Data points show responses of individual zebrafish before light exposure and afterwards at the time points indicated (x-axis). Bars show group mean ± 95% CI. ****p<0.0001, 2-way ANOVA with Šidák multiple comparisons test. (**B**) Heart rate (y-axis) was quantified in the same experimental groups and time points as panel A. Data points show heart rates of individual zebrafish, bars show group mean ± 95% CI. (**C**) Degenerating cells in MG2I-treated NeuMitoFAP zebrafish were labeled with acridine orange, before, and 2 and 24 hr after exposure to 60 J/cm² far-red light. The images show confocal z-plane projections through the spinal cord of immobilized live zebrafish larvae. dL5**-mCerulean3 is pseudocolored blue and acridine orange-labeled cells yellow. (**D**) Acridine orange-labeled cells in the spinal cord of live zebrafish were counted (y-axis), before far-red light exposure and at the indicated time points afterwards (x-axis). Experimental groups: WT zebrafish treated with MG2I (green), NeuMitoFAP zebrafish (blue), NeuMitoFAP zebrafish treated with MG2I (red). Data points show individual zebrafish, bars show mean ± SE. *p<0.05, ****p<0.0001, NeuMitoFAP-MG2I zebrafish versus other groups at the same time point; #p<0.05, ####p<0.0001, NeuMitoFAP-MG2I zebrafish at the indicated time point versus pre-exposure value; 2-way ANOVA with Tukey multiple comparisons test. (**E**) Transmission electron micrographs of sections from the telencephalon of NeuMitoFAP zebrafish treated with MG2I before, and at the indicated time points after, far-red light exposure. The upper image of each pair shows a low-magnification view, and the lower image shows a high-magnification view illustrating ultrastructural features.

The online version of this article includes the following source data and figure supplement(s) for figure 6:

**Source data 1.** Source data for *Figure 6D*.
**Figure supplement 1.** Delayed CNS cell death in NeuMitoFAP zebrafish exposed to MG2I and far-red light.

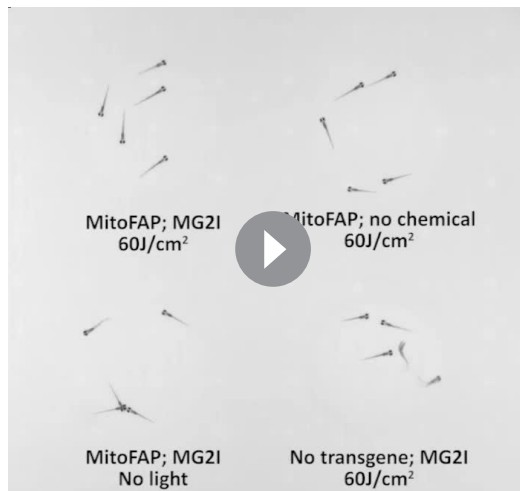

**Video 3.** Persistent motor deficits in NeuMitoFAP-MG2I zebrafish following far-red light exposure. 7dpf zebrafish are shown 48 hr after far-red light exposure at 5dpf. Methods and experimental groups are identical to *Video 1*. The severe neurological abnormalities seen immediately after far-red light exposure at 5dpf persist in the NeuMitoFAP-MG2I-light group after two days of recovery.

https://elifesciences.org/articles/51845#video3

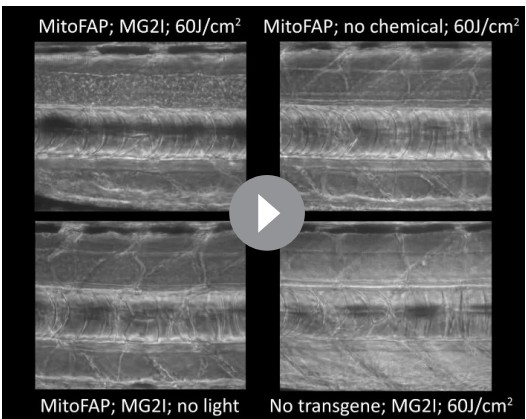

**Video 4.** Normal circulatory and cardiac development in NeuMitoFAP-MG2I zebrafish following far-red light exposure. Phase contrast videomicrography of 7dpf zebrafish 48 hr after far-red light exposure at 5dpf. Methods and experimental groups are identical to *Video 2*. Despite the persistence of severe neurological abnormalities two days after light exposure, circulation and heartbeat in the NeuMitoFAP-MG2I-light group remain normal, showing that irreversible abnormalities are restricted to the nervous system.

https://elifesciences.org/articles/51845#video4

$10.1 \pm 0.6$; NeuMitoFAP-MG2I, $18.5 \pm 2.1$ AO-labeled cells/spinal cord; mean $\pm$ SE). Immediately following far-red light exposure, cell death started to increase steadily in NeuMitoFAP-MG2I larvae but not controls, peaking at 24 hr post-exposure. At this time point, over 30-fold more AO-labeled cells were found in NeuMito-FAP-MG2I larvae than controls (WT-MG2I, $9.7 \pm 0.8$; NeuMitoFAP, $9.7 \pm 1.1$; NeuMitoFAP-MG2I, $311.7 \pm 14.8$ AO cells/spinal cord; $p<10^{-15}$ NeuMitoFAP-MG2I vs. NeuMitoFAP, $p<10^{-15}$ NeuMitoFAP-MG2I vs. MG2I, $p<10^{-15}$ NeuMitoFAP-MG2I at 24 hr vs. baseline, 2-way ANOVA with Tukey multiple comparisons test; *Figure 6C, D*). Transmission electron microscopy was employed to examine the underlying ultra-structural changes (*Figure 6E*). Prior to far-red light exposure, NeuMitoFAP-MG2I neurons showed normal morphology with homogenous nuclei, prominent nucleoli and elongated mito-chondria with densely packed cristae. Immediately after far-red light exposure, severe and widepread mitochondrial abnormalities were observed in neurons, as described above. Remarkably, however, despite these dramatic mitochondrial changes, the adjacent nuclear membrane, Golgi apparatus and endoplasmic reticulum appeared normal at this time point. By 2 hr post-exposure, scattered neurons started to show nuclear chromatin condensation and other morphological changes, including nuclear mem-brane separation. At 24 hr post-exposure, numerous apoptotic bodies were visible, along with neurons showing morphological features suggesting ongoing apoptosis, and other neu-rons showing signs suggestive of necrosis.

## Discussion

We have demonstrated dramatic changes in the neurological function of an intact living verte-brate organism following chemoptogenetic tar-geting of mitochondrial function with organelle-level spatial precision. Our data show a direct link between neuronal respiration, bioenergetics and physiology; quantify neuronal respiration and its bioenergetic contributions in vivo; and demonstrate that neuronal death following mito-chondrial damage is delayed and presumably dependent on secondary mechanisms. The new transgenic lines and methods we report will pro-vide powerful tools for investigating mitochon-drial homeostasis and pathophysiology, and for understanding the neural basis of behavior.

The requirement of all three components of the chemoptogenetic system (dL5**, MG2I and

far-red light) shows that the observed phenotypes were caused by singlet oxygen. The subcellular localization of the dL5**-mCerulean3 fusion protein in NeuMitoFAP zebrafish resulted in spatially-restricted $^1O_2$ production within neuronal mitochondria. Correspondingly, initial damage was confined to mitochondria, as evidenced by preservation of cellular ultrastructure immediately adjacent to severely damaged mitochondria directly after light exposure. Potentially, $^1O_2$ can react with all mitochondrial macromolecules and the biochemical targets of this reactive oxygen species in the mitochondrion are not yet fully resolved. In cultured HEK293 cells expressing mitochondrially-targeted dL5**-mCerulean3, exposure to MG2I and far-red light decreased the activity of mitochondrial respiratory chain complexes I, III and IV (*Qian et al., 2019*). COX4 and COX8 encode subunits of complex IV (CIV, cytochrome C oxidase), and it is possible that the fusion protein is directed to CIV by the COX8 sequence remaining after cleavage of the signal peptide. In this case, localized $^1O_2$ damage to the CI-CIII$_2$-CIV super-complex ('respirasome') (*Letts and Sazanov, 2017*) is predicted to decrease electron transport and thereby dissipate the inner membrane proton gradient, resulting in loss of ATP production and mitochondrial swelling. Although this would account for the loss of respiration and bioenergetic collapse we observed, it is unclear whether respiratory chain function is disrupted by $^1O_2$ directly. In HEK293 cells expressing mitochondrially-targeted dL5**-mCerulean3, electron transport and respiration following light exposure were more severely disrupted by a secondary wave of ROS produced by damaged mitochondria (*Qian et al., 2019*). Consequently, the presence of oxidative damage does not necessarily imply that a molecule is a primary target of $^1O_2$. Furthermore, the severe ultrastructural changes we observed immediately after light exposure raise the possibility that changes in mitochondrial function could occur as a consequence of disrupted mitochondrial architecture. Mitochondrial cristae are dynamic structures whose organization strongly influences assembly of respiratory chain super-complexes and electron transport function (*Cogliati et al., 2016*). Primary $^1O_2$-induced damage to proteins involved in forming or regulating cristae, for example OPA1 (*Frezza et al., 2006*; *Patten et al., 2014*) or components of the MICOS complex (*Kozjak-Pavlovic, 2017*), or direct damage to inner membrane lipids, could cause indirect changes in respiratory function by disrupting the topological organization of electron transport chain complexes in the inner mitochondrial membrane (*Cogliati et al., 2013*). Additional studies will be necessary to determine the initial, direct targets of $^1O_2$ in this model.

FAP-MG2I zebrafish showed a large decrease in whole-animal respiration following exposure to far-red light. The 12 kb *eno2* regulatory element is expressed in the majority of CNS and PNS neurons, most strongly in large projection neurons such as retinal ganglion cells, reticulospinal neurons and motor neurons, that are predicted to be most metabolically active (*Bai et al., 2007*; *Bai et al., 2009*). If it is presumed that light exposure fully disrupted cellular respiration in NeuMitoFAP-MG2I cells, it can be inferred that *eno2*-expressing neurons account for nearly 25% of the baseline $O_2$ consumption in an immobilized zebrafish larva. The dramatic loss of respiratory activity following light exposure resulted in profound bioenergetic consequences. The overall 10% decrease in whole-animal ATP levels suggests a much larger ATP deficit in *eno2*-expressing neurons that comprise a relatively small fraction of the total cells in a larval zebrafish. It has been estimated that up to 50% energy expenditure in neurons is devoted to maintaining the transmembrane ionic gradients underlying the resting membrane potential (*Howarth et al., 2012*). Progressive depolarization of NeuMitoFAP-MG2I neurons during light exposure was rescued by phosphocreatine, a substrate for creatine kinase (CK) that allows ATP generation from ADP, independent of oxidative phosphorylation. This observation formally establishes that loss of neural function in this model was attributable to bioenergetic crisis and highlights the importance of mitochondrial respiration for providing the ATP necessary for physiological functions such as active ion transport in neurons. In addition, these findings provide further support to the specificity of the initial oxidative insult. A cysteine residue within the active site of CK is necessary for enzymatic activity, which has been shown in other experimental systems to be readily inactivated by oxidants including dopamine (*Van Laar et al., 2008*), superoxide (*Yuan et al., 1992*) and peroxynitrite (*Konorev et al., 1998*). The preservation of CK function in NeuMitoFAP zebrafish neurons sufficient to allow rescue of cellular bioenergetics by phosphocreatine suggests that $^1O_2$-mediated damage did not inactivate cytoplasmic CK isoforms, even in the presence of severe mitochondrial disruption.

The persistence of acute neurobehavioral deficits in the days following light exposure was attributable to cell death. This was maximal 24 hr after light exposure, well after acute neurobehavioral changes were first observed. We predict that the initial mitochondrial damage provoked delayed

loss of cellular viability through secondary mechanisms, several of which may be involved. First, ATP depletion is a well-recognized cause of cellular necrosis. In this regard, the reliance of neuronal ATP synthesis on oxidative phosphorylation may be a critical factor distinguishing neurons from cultured cells, which showed compensatory glycolysis and cell cycle arrest, but not loss of viability, following dL5**-MG2I-induced mitochondrial damage (*Qian et al., 2019*). Second, it is likely that cellular $Ca^{2+}$ homeostasis was disrupted in NeuMitoFAP-MG2I neurons following far-red light exposure. Cytosolic $Ca^{2+}$ levels are maintained by ATP-dependent transport of $Ca^{2+}$ into the endoplasmic reticulum and extracellular space, and by mitochondrial $Ca^{2+}$ buffering that is dependent on the inner mitochondrial membrane electrochemical potential and its structural integrity. Each of these processes is likely to have been impaired in NeuMitoFAP-MG2I neurons following light exposure. Depending on severity and subcellular distribution, elevated cytosolic $Ca^{2+}$ can trigger either necrotic or apoptotic cell death (*Pinton et al., 2008*). Finally, compromised mitochondrial structure may allow release of pro-apoptotic mediators. For example, cytochrome *c* is retained physiologically within cristae folds; consequently, the obliteration of mitochondrial cristae observed in this model is predicted to allow cytochrome *c* efflux into the cytoplasm, where it can initiate *Apaf-1*-dependent apoptosome assembly and activation of the intrinsic apoptotic pathway (*Kroemer et al., 2007*). Since morphological markers of both apoptosis and necrosis were visible, it seems unlikely that a single downstream mechanism was responsible for the observed cell death. The bioenergetic requirements of individual neurons vary according to their morphology and activity. Further, dL5** expression varies between different neuronal populations in NeuMitoFAP zebrafish, because the *eno2* regulatory element is expressed differentially in discrete cell types (*Bai et al., 2007*; *Bai et al., 2009*). We predict that the mechanisms by which neurons die in this model depend on the relationship between bioenergetic demand and loss of ATP, similar to other cell types (*LiebERThal et al., 1998*). For example, a cell with high ATP requirements that sustains a severe mitochondrial injury might undergo necrotic cell death rapidly, whereas less prominent damage in a cell with more modest ATP demands might trigger signaling events culminating in delayed programmed cell death.

Targeting neuronal mitochondrial function with dL5**-MG2I provides new experimental opportunities in vivo, and the tools reported here will be useful for multiple downstream applications. Use of Gal4-UAS genetics will greatly expand the utility of the approach, because the dL5**-mCerulean3 fusion protein can be expressed in the mitochondria of any cell type in vivo, by crossing Tg(*UAS: COX4-COX8-dL5**-mCer3*)[pt427] zebrafish to a tissue-specific Gal4 driver line. Many relevant transgenic driver lines are available that express Gal4 in dopaminergic neurons (*Fujimoto et al., 2011*) or glial cells (*Frøyset et al., 2018*) of interest to disease pathogenesis, in addition to larger libraries of enhancer trap Gal4 insertions useful for functional neuroanatomy studies (*Bergeron et al., 2015*; *Kawakami et al., 2010*; *Marquart et al., 2015*). Furthermore, by exploiting the dependence of neurons on oxidative phosphorylation, our approach provides several improvements on current technology for targeted cell ablation to analyze neural circuits underlying behavior (*Bergeron et al., 2015*; *Godoy et al., 2015*) The most widely applied method for chemogenetic ablation in zebrafish models relies on metronidazole-induced DNA crosslinking in transgenic animals expressing the bacterial enzyme nitroreductase (*Curado et al., 2007*). DNA damage in this model takes many hours, and sometimes days, to accumulate to a level sufficient to ablate the targeted cell groups. In contrast, the rapid light-induced neuronal depolarization caused by mitochondrially-targeted dL5**-MG2I provides temporal certainty regarding lesion onset, avoids pharmacokinetic uncertainties inherent in chemical approaches, and occurs with sufficient rapidity to prevent compensatory changes in circuitry that may obscure the resulting neurobehavioral consequences. Furthermore, spatial specificity can be enhanced by directing the activating light to particular neurons or subcellular regions of interest, for example by using a far-red laser beam rather than a diffused light source to excite dL5**-MG2I. This will be of value for future investigations into the compartmentalization of bioenergetics and maintenance of functional mitochondrial biomass in neurons in vivo. Illuminating individual topological domains of NeuMitoFAP neurons may allow direct exploration of how mitochondrial spatial distribution contributes to maintaining bioenergetic requirements and ionic gradients throughout the dendritic arborization, cell body and axonal projection. In addition, mitochondrial quality control has been implicated in the pathophysiology of neurological diseases including Parkinson's disease, but the use of chemicals that non-selectively depolarize all mitochondria simultaneously precludes investigation of the underlying mechanisms in specific neurons. dL5**-MG2I provides several advantages over other genetically-encoded photosensitizers such as KillerRed (*Bulina et al., 2006*) for these

applications, including minimal photobleaching, high quantal yield of $^1O_2$, and excitation by tissue-penetrant far-red light (*He et al., 2016*). In addition, the necessity for addition of a chemical fluorogen to photosensitize FAP transgenic animals circumvents the practical challenges inherent in generating and propagating transgenic animals expressing constitutively active photosensitizers, which must be raised and handled in the dark.

Precision subcellular ablation is likely to have broad applications in neuroscience beyond investigation of mitochondrial function and zebrafish models. Future development of this approach will include generation of constructs that direct dL5** to other cellular components, such as nuclear subdomains, lysosomal proteins or key components of pre- or post-synaptic terminals, allowing resolution of their specific contributions to cellular physiology and pathophysiology. In addition, the system is fully portable to other experimental systems, including mammalian models.

## Materials and methods

### DNA constructs

pT2KSAGFF (*Asakawa et al., 2008*) and pT2-5UASMCS (*Asakawa et al., 2008*) were gifts from Dr. Koichi Kawakami, National Institute of Genetics, Tokyo, Japan. To generate the driver construct, a 325 bp *Eco*RI/*Bgl*II restriction fragment encoding Gal4FF was released from pT2KSAGFF and, after Klenow blunting the *Bgl*II overhang, ligated into the *Eco*RI and Klenow-blunted *Pac*I sites of pBS-I-Sce1-GFP-eno2-5′−3′-arm (*Bai et al., 2007*). The resulting plasmid was linearized with *Hin*dIII, and the 12 kb *eno2* regulatory sequence, encompassing exons 1 and 2 and genomic flanking region, was captured from BAC zC51M24 by gap repair recombination, as described in our prior work (*Bai et al., 2007*) to yield pBS-I-Sce1-eno2:Gal4FF. To generate the responder construct, a 1.74 kb *Pme*I/*Xho*I restriction fragment was released from pcDNA3.1-cox4-cox8-dL5-2xG4S-mCerulean3 (*Qian et al., 2019*; *Telmer et al., 2015*) (Addgene 73208) and ligated into the *Xho*I and Klenow-blunted *Eco*R1 sites of pT2-5UASMCS to yield pTol2-5UAS:cox4-cox8-dL5-2xG4S-mCerulean3 (2xG4S encodes a flexible GGGGSGGGGS linker between dL5** and mCerulean3 to allow correct folding of both protein domains). Both plasmids were verified by DNA sequencing and restriction digest.

### Zebrafish

Zebrafish embryos were raised in E3 buffer (5 mM NaCl, 0.17 mM KCl, 0.33 mM CaCl$_2$, 0.33 mM MgSO4; unless otherwise stated, all chemicals were supplied by Sigma, St. Louis, MO) at 28.5°C, under cyclic illumination comprising 14 hr green light:10 hr dark. Transgenic zebrafish were generated as described in our previous work, using the I-*Sce*1 meganuclease method (*Bai et al., 2007*) for Tg(*eno2:gal4ff*) and the Tol2 transposon method (*Dukes et al., 2016*) for Tg(*UAS:COX4-COX8-dL5**-mCer3*). Multiple transgenic F1 founders were identified by PCR genotyping (primer sequences: eno2-GFF-F, 5′-GTCTTCTATCGAACAAGCATGC-3′; eno2-GFF-R, 5′-CATGTCAAGGTCTTC TCGAGG-3′; mitoFAP-F, 5′-CCGTCGTTACCCAAGAACC-3′; mitoFAP-R, 5′-TCCTGAGTCACCA-CAGCC-3′) and lines were selected for analysis based on robust transgene expression and minimal variegation. The lines reported here show Mendelian inheritance of single transgene insertions and have been assigned allele designations Tg(*eno2:gal4ff*)[pt425] and Tg(*UAS:COX4-COX8-dL5**-mCer3*)[pt427]. Analyses were carried out using F3 and later generations after backcrosses to WT zebrafish. A minimum of three independent biological replicates were completed for all experiments.

### Microscopy

Double transgenic NeuMitoFAP animals were identified by epifluorescence microscopy for the mCerulean3 reporter. Tricaine-anesthetized larvae were positioned in 3% methylcellulose for acquisition of live epifluorescence images using an Olympus MVX-10 stereo zoom microscope and SPOT camera (Olympus, Center Valley, PA). Confocal images of tricaine-anesthetized larvae, mounted in low melting point agarose in contact with the coverslip glass of a MatTek dish (MatTek Corporation, Ashland, MA), were acquired using an Olympus IX-81 inverted microscope and Fluoview confocal system (Olympus). Acridine Orange-labeled neurons were visualized using an inverted epifluorescence microscope (Olympus IX-71) and counted manually or imaged by confocal microscopy as

above. For intravital imaging of mitochondria, zebrafish larvae were anesthetized in tricaine, exposed to far-red light if appropriate and then mounted on their sides in low melting point agarose at the bottom of a Mattek dish, so that the skin was in contact with the glass coverslip allowing visualization of the lateral line nerve. Images were acquired using a Leica SP8 confocal microscope with HC PL APO 1.30 NA 93x glycerol-immersion objective (Leica Microsystems, Buffalo Grove, IL). mCer3 was excited at 405 nm and emitted light collected from 447 to 699 nm with temporal gating between 0.6–6 ns using a Leica Acousto-Optical Beam Splitter. Images were taken with a 0.600 ms dwell time and 4x line averaging. Images were analyzed using NIS-Elements (Nikon Instruments, Melville, NY), by reducing Z-stacks to 2D images using extended depth focus, then binarizing the resulting images to show regions of CFP fluorescence, allowing automated measurements of circularity, size and number.

## Immunofluorescence

Larvae were fixed in 4% paraformaldehyde in PBS at 4°C for 4 hr. 14μm-thick cryosections were incubated with primary antibody (chicken anti-GFP #ab13970, 1:5000, Abcam, Cambridge, MA; rabbit anti-TOM20 sc-11415, 1:1000, Santa Cruz, Dallas, TX) at 4°C for 16 hr, washed three times in PBS and incubated in secondary antibody (Alexa 488 goat anti-chicken A11039, 1:10000, and Alexa 555 goat anti-rabbit A11034, 1:10000, Thermo Fisher, Waltham, MA). After three further washes, sections were incubated in DAPI (200 ng/mL in PBS) and mounted in Gelmount (Electron Microscopy Sciences, Hatfield, PA) for confocal imaging as detailed above. Colocalization analysis was carried out using Olympus FluoView software.

## Chemoptogenetic ablation

500 nM MG2I (synthesized as described previously *He et al., 2016*) was added to larval E3 buffer at 72hpf (hours post fertilization). After treatment with MG2I, NeuMitoFAP zebrafish were housed and handled under green LED safelight illumination (LSM-G3 × 3, SuperBrightLEDs, St. Louis, MO). Tricaine-anesthetized zebrafish were exposed to far-red light in 35 mm Mat-Tek dishes at 5dpf. A Chanzon 100W Deep Red LED array (Amazon, Seattle, WA) was mounted on a heatsink and fan and suspended from a custom-built light stand (*Figure 1—figure supplement 1*), allowing the 35 mm dish to be illuminated from below without heat transfer to the water (*Figure 3—figure supplement 1*). The LED power circuit was controlled using a microcontroller board (Arduino Uno, Amazon, Seattle, WA) for precise timing of exposure and adjustable power up to 160 mW/cm$^2$ (*Figure 1—figure supplement 1*). The emission spectra of all light sources were verified using a spectrometer (BLK-CXR, Stellarnet, Tampa, FL).

## Neurobehavioral analysis

Locomotor function was analyzed as reported in our prior work (*Zhou et al., 2014*). Larvae were transferred to 96 well plates at 5dpf under green LED illumination using a large-bore Pasteur pipette with a flame-polished aperture, then acclimatized to the recording chamber for 30 min at 28.5°C. The visual motor response was elicited using green light (*Burton et al., 2017*) and recorded with a USB 3.0 camera (#FL3-U3-13Y3M-C, Point Gray Research, Richmond, BC, Canada) under infrared illumination (#BL812-880, Spectrum Illumination, Montague, MI). Video recordings were analyzed offline using our published open-source MATLAB applications *LSRtrack* and *LSRanalyze* (*Cario et al., 2011*). All data were derived from recordings with <5% total tracking errors. Heart rates of tricaine-anesthetized zebrafish were counted manually by direct visualization of cardiac contractions through a dissecting microscope.

## Electrophysiology

Larvae were anesthetized in 0.02% tricaine, paralyzed by exposure to 1 mg/mL α-bungarotoxin for 10 min, then secured through the notochord to a layer of silicone (Sylguard, Corning) in the bottom of a 35 mm culture dish, using 0.025 mm tungsten pins. The preparation was continuously superfused with extracellular solution (NaCl 131 mM, KCl 2 mM, $KH_2PO_4$ 1.25 mM, $NaHCO_3$ 20 mM, $MgCl_2$ 2 mM, $CaCl_2$ 2.5 mM, Glucose 10 mM, bubbled with 95% oxygen/5% $CO_2$, pH 7.4) at rate of 2 mL/min throughout recording. A flap of skin was reflected using a fine tungsten probe to expose the posterior lateral line ganglion. The region was imaged during recording using an Olympus

BX51WI microscope with 40x water immersion objective and infrared DIC optics, illuminated with an Olympus U-LH100 IR halogen light source with 32BP775 IR bandpass filter (Olympus, Center Valley, PA), and visualized using a USB 3.0 video camera (#FL3-U3-20E4M-C, Point Gray Research, Richmond, BC, Canada). Glass microelectrodes with resistance of 6–10 MΩ were filled with intracellular solution (KGlu 126 mM, KCl 15 mM, NaCl 10 mM, HEPES 10 mM, MgCl$_2$ 2 mM, pH 7.2; phosphocreatine 10 mM was added for one experimental group) and connected to a headstage (HS-9A $\times$ 0.1 U, Molecular Devices, San Jose, CA) mounted on a motorized micromanipulator (MP-225, Sutter Instrument, Novato, CA). Patch clamp recordings were made from posterior lateral line ganglion neurons in a whole-cell configuration, using a microelectrode amplifier (Axoclamp 900A, Molecular Devices, San Jose, CA) in current clamp mode. Signals were digitized at 4 kHz sampling rate (Digidata 1440A, Molecular Devices). A far-red LED light source (Chanzon 100W Deep Red LED array, Amazon, Seattle, WA) was mounted on a heatsink under the recording chamber. Baseline membrane potential was recorded for 10 min, following which the LED was activated and recording continued for a further 20–30 min. Data were analyzed offline (Clampfit 10.7 module of pCLAMP, Molecular Devices).

## Respiration

Oxygen consumption rate (OCR) and bath acidification rate (BAR) were measured using a Seahorse XF24 Extracellular Flux Analyzer (Agilent, Santa Clara, CA). Zebrafish larvae were anesthetized in 0.015% tricaine and then a small cut made in the distal caudal fin to facilitate absorption of drugs from the bath. After incubation in 40 µM d-tubocurarine for 5–10 min until spontaneous and touch-evoked muscle contractions were lost, larvae were transferred into the wells of a Seahorse XF24 Islet microplate (Agilent, Santa Clara, CA) containing 800 µL of E3 buffer, positioned in the center of each well using an eyelash, and then secured by islet plate capture screens. Four buffer-only wells on each plate served as negative controls. After sensor calibration, basal OCR and BAR were quantified over 24 cycles, each consisting of 1.5 min mixing and 4 min measurement. For measurements on dissociated cells, freshly-dissected adult NeuMitoFAP zebrafish brains were incubated with 0.25% trypsin/EDTA (Thermo Fisher, Waltham, MA) at 37°C for 20 min, which was inactivated by adding 5 volumes of 10% fetal bovine serum (FBS) in PBS. Samples were pipetted to dissociate cells, centrifuged at 2000 rpm, washed with 2% FBS in PBS and passed through a 40 µm cell strainer (BD Falcon, Corning, NY), following which cells were incubated with 100 nm MG2I in the dark for 20 min at 20°C then washed and exposed to far-red light using the same apparatus as used for zebrafish. $3 \times 10^5$ cells in DMEM (# D7777, Sigma, St, Louis, MO) were plated into each well of a 96-well Seahorse Xfe96 FluxPak plate (Agilent, Santa Clara, CA) and equilibrated at 28°C for one hour prior to running the Seahorse assay. Inhibitors and final concentrations were: Oligomycin (1 µg/mL; Sigma, #O4876), FCCP (300 nM; Sigma, #C2920), rotenone (1 µM; Sigma, R8875), antimycin-A (1µM; Sigma, A8674).

## ATP quantification

2 hr after exposure to far-red light, groups of 5 larvae were collected and homogenized in 50 µL lysis buffer (tris 30 mM, urea 9M, CHAPS 2% w/v, pH 7.4) on ice, then centrifuged at 10,000 g for 15 min. ATP concentration was measured in the supernatant using a bioluminescence assay (ATP Determination Kit, Invitrogen, Carlsbad, CA) and a luminometer (LMax II, Molecular Devices, San Jose, CA), in comparison with a reference curve of known ATP concentrations. Values for each sample were then normalized to protein concentration, measured using a Bradford assay (BioRad, Hercules, CA) and microplate reader (SPECTRAmax PLUS384, Molecular Devices, San Jose, CA) in comparison with a reference curve of known bovine albumin concentrations.

## Acridine Orange labeling

A 100 mg/mL stock solution of AO in ethanol was stored at 4°C and diluted in E3 buffer immediately before use to a final concentration of 5 µg/mL. Zebrafish larvae were incubated in the resulting solution at 28°C for 30 min in the dark, washed twice for 10 min each in E3 buffer, anesthetized in 0.015% tricaine, and mounted in low melting point agarose against the coverslip glass of a MatTek dish for imaging.

## Electron microscopy

Zebrafish larvae were fixed in Karnovsky fixative at 4°C for 16 hr, the skull and surrounding tissues dissected to expose the brain, then fixed for a further 24 hr. Samples were then rinsed in PBS, post-fixed in 1% osmium tetroxide with 1% potassium ferricyanide, rinsed in PBS, dehydrated through a graded series of ethanol and propylene oxide and embedded in Poly/Bed 812 (Polysciences, Warrington, PA). 300nm-thick sections were stained with 0.5% Toluidine Blue in 1% sodium borate and examined under a light microscope to identify regions of interest. 65 nm-thick sections were stained with uranyl acetate and Reynold's lead citrate, then imaged using a JEOL 1011 transmission electron microscope with a side mount AMT 2K digital camera (Advanced Microscopy Techniques, Danvers, MA).

## Experimental design and data analysis

All experiments were repeated in at least three independent biological replicates (different cohorts of larval zebrafish bred on different days). Data from replicate experiments whose results did not differ significantly were combined for clarity of presentation, as indicted in individual figure legends. Experimental groups were defined firstly by genotype, then larvae allocated randomly to receive MG2I or no chemical.

## Acknowledgements

This work was supported by NIH research grants ES025606, ES022644, RR019003, and EB017268. We thank the aquatics team at the University of Pittsburgh Department of Laboratory Animal Resources for expert care of our transgenic zebrafish lines. We thank Travis Wheeler (Department of Cell Biology and Physiology, University of Pittsburgh) for construction of the light stand and neurobehavioral equipment. WX and BJ are Tsinghua University Scholars at the University of Pittsburgh and are supported by the China Scholarship Council.

## Additional information

### Funding

| Funder | Grant reference number | Author |
|---|---|---|
| National Institute of Environmental Health Sciences | ES025606 | Qing Bai<br>Dmytro Kolodieznyi<br>Patricia L Opresko<br>Claudette M St Croix<br>Simon Watkins<br>Bennett Van Houten<br>Marcel P Bruchez<br>Edward A Burton |
| National Institute of Environmental Health Sciences | ES022644 | Qing Bai<br>Vladimir A Ilin<br>Claudette M St. Croix<br>Edward A Burton |
| National Center for Research Resources | RR019003 | Ming Sun<br>Donna B Stolz<br>Claudette M St. Croix<br>Simon Watkins |
| National Institute of Biomedical Imaging and Bioengineering | EB017268 | Ming Sun<br>Donna B Stolz<br>Claudette M St Croix<br>Simon Watkins<br>Marcel P Bruchez |
| China Scholarship Council | | Wenting Xie<br>Binxuan Jiao |

The funders had no role in study design, data collection and interpretation, or the decision to submit the work for publication.

## Author contributions

Wenting Xie, Binxuan Jiao, Investigation, Methodology, Designed and carried out experiments, collected, analyzed and interpreted data, contributed to writing the manuscript; Qing Bai, Conceptualization, Resources, Supervision, Investigation, Methodology, Generated novel transgenic zebrafish lines, designed and carried out experiments, collected, analyzed and interpreted data, contributed to writing the manuscript; Vladimir A Ilin, Conceptualization, Software, Investigation, Methodology, Designed and carried out experiments, collected, analyzed and interpreted data, contributed to writing the manuscript; Ming Sun, Investigation, Prepared samples and carried out electron microscopy; Charles E Burton, Software, Formal analysis, Developed analysis software for zebrafish behavioral experiments; Dmytro Kolodieznyi, Resources, Synthesized MG2I used in the study; Michael J Calderon, Investigation, Methodology; Donna B Stolz, Formal analysis, Investigation, Methodology, Supervised and interpreted electron microscopy; Patricia L Opresko, Conceptualization, Funding acquisition, Methodology, Project administration; Claudette M St Croix, Conceptualization, Resources, Investigation, Visualization, Methodology; Simon Watkins, Conceptualization, Resources, Formal analysis, Investigation, Methodology; Bennett Van Houten, Conceptualization, Resources, Funding acquisition, Project administration; Marcel P Bruchez, Conceptualization, Resources, Formal analysis, Funding acquisition, Investigation, Methodology; Edward A Burton, Conceptualization, Resources, Data curation, Software, Formal analysis, Supervision, Funding acquisition, Investigation, Methodology, Project administration, Built equipment, designed and carried out experiments, analyzed and interpreted data, wrote the manuscript

## Author ORCIDs

Dmytro Kolodieznyi (iD) http://orcid.org/0000-0001-5272-6856
Simon Watkins (iD) http://orcid.org/0000-0003-4092-1552
Marcel P Bruchez (iD) http://orcid.org/0000-0002-7370-4848
Edward A Burton (iD) https://orcid.org/0000-0002-8072-4636

## Ethics

Animal experimentation: Experiments were carried out in accordance with the NIH Guide for the Care and Use of Laboratory Animals and were approved by the University of Pittsburgh Institutional Animal Care and Use Committee (IACUC protocol #17019974; PHS assurance #D16-00118) as meeting standards for humane animal care.

## Decision letter and Author response

Decision letter https://doi.org/10.7554/eLife.51845.sa1
Author response https://doi.org/10.7554/eLife.51845.sa2

# Additional files

## Supplementary files

• Transparent reporting form

## Data availability

All data generated or analysed during this study are included in the manuscript and supporting files.

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
