## [Decision Letter]

**Acceptance summary:**

This manuscript provides a thorough characterization of a new, powerful, tool to assess the impact of impaired mitochondrial function on neurons and in an accessible in vivo vertebrate model system. The ability to use light to damage organelles in a tissue-specific manner will be useful for all interested in mitochondrial biology.

**Decision letter after peer review:**

Thank you for submitting your article "Chemoptogenetic ablation of neuronal mitochondria in vivo with spatiotemporal precision and controllable severity" for consideration by *eLife*. Your article has been reviewed by three peer reviewers, including Stephen C Ekker as the Reviewing Editor and Reviewer #1, and the evaluation has been overseen by Didier Stainier as the Senior Editor. The following individual involved in review of your submission has agreed to reveal their identity: Jeff S Mumm (Reviewer #2).

The reviewers have discussed the reviews with one another and the Reviewing Editor has drafted this decision to help you prepare a revised submission.

Summary:

The authors describe an innovative chemoptogenetic approach for disrupting mitochondrial function in targeted neurons in vivo. Xie et al. have used transgenesis combined with novel chemoptogenetic techniques to create a zebrafish line that can be used to systematically induce oxidative stress in mitochondria of various cell types using tissue specific Gal4 drivers. This line will be incredibly useful to the mitochondrial biology community as it will enable researchers to test the role of mitochondrial oxidative stress in disease etiology, a major open question in the field. The authors have gone to great lengths to show the basic system and the complex effects they see are not due to some side-effects or other technical artifacts. There are some concerns, however, about the true mediators of their highly localized induction of activated oxygen. This is appropriate as a resource paper for *eLife*.

Essential revisions:

1) Adjustable damage: The authors state, "severity and rate of induction can be regulated by adjusting light exposure time and power", which was demonstrated for locomotor activity inhibition. Given the potential for providing insight into mechanisms regulating reparative responses to mitochondrial damage and ATP depletion in neurons, an example of sub-lethal induction, i.e., sufficient for a quantifiable change without incurring cell ablation, would provide a valuable demonstration of this property that would further support the adjustable nature of the tool and substantially enhance the impact of the manuscript.

2) It is unclear if defects to the mitochondria are the cause of the neuronal activity alterations and cell death. Clearly the organelles do not look healthy by TEM but the effect of singlet oxygen on mitochondrial health and function are not tested. Since the mitochondria are already labeled with a fluorescent protein, it would be relatively simple to assess mitochondrial localization and shape after the insult.

3) The EM analyses were not well described. Such work – even with the extreme dysmorphology reported here – needs to be assessed in a blinded fashion. The authors could take the assembled scans, encode them, and have a blinded assessment to increase rigor of this work.

4) Statistical analysis: The 1-way and/or 2-way ANOVA used to evaluate data in Figures 2D, 3B-C, 5A, 5C-D, and Figure 2—figure supplement 2B is not sufficient to assess differences between groups. A post hoc analysis with correction for multiple comparisons is required, as per Figure 2—figure supplement 2B, Figure 2—figure supplement 3B-C and Figure 4—figure supplement 1B. Figure legends should indicate if dots in scatter plots represent individual fish of a single exemplary experiment or across all replicates.

---

## [Author Response]

Essential revisions:1) Adjustable damage: The authors state, "severity and rate of induction can be regulated by adjusting light exposure time and power", which was demonstrated for locomotor activity inhibition. Given the potential for providing insight into mechanisms regulating reparative responses to mitochondrial damage and ATP depletion in neurons, an example of sub-lethal induction, i.e., sufficient for a quantifiable change without incurring cell ablation, would provide a valuable demonstration of this property that would further support the adjustable nature of the tool and substantially enhance the impact of the manuscript.

We previously showed that the severity of the acute neurological phenotype was dependent on the amount of far-red light energy to which the zebrafish were exposed (Figure 2—figure supplement 2). We now show further evidence of the adjustable nature of our approach and the potential for induction of sub-lethal damage: (i) we have measured motor function during far-red light exposure, showing that the earliest abnormality is hyperkinesia, which advances to progressive hypokinesia as far-red light illumination is continued and cumulative energy exposure increases (new Figure 2—figure supplement 4); (ii) we show evidence of a dose-response relationship between far-red light exposure and electrophysiological abnormalities, including non-lethal partial depolarization after transient far-red light exposure (new Figure 3—figure supplement 2); (iii) we show a dose-response relationship between far-red light exposure and mitochondrial respiratory function in dissociated brain cells by Seahorse assay (new Figure 4F and G). Together these data show unequivocally that the severity of mitochondrial damage and the resulting phenotypes are adjustable by far-red light dose.

2) It is unclear if defects to the mitochondria are the cause of the neuronal activity alterations and cell death. Clearly the organelles do not look healthy by TEM but the effect of singlet oxygen on mitochondrial health and function are not tested. Since the mitochondria are already labeled with a fluorescent protein, it would be relatively simple to assess mitochondrial localization and shape after the insult.

Rescue of electrophysiological abnormalities by phosphocreatine (Figure 3) formally proves that neuronal depolarization was caused by bioenergetic collapse, which is supported by the decreased ATP levels we measured in larvae after light exposure (Figure 4A). In neurons, bioenergetic crisis characteristically results from loss of mitochondrial function. The involvement of singlet oxygen in causing these abnormalities is strongly supported by the absolute requirement of all three components of the chemoptogenetic system (FAP, MG2I and far-red light) to provoke relevant phenotypic changes (Figures 2-6); formal proof that the deleterious cellular effects of FAP/MG2I/far-red light are attributable to singlet oxygen was shown in prior publications^1, 2^. The effect of singlet oxygen on mitochondrial health is shown as (i) loss of mitochondrial respiratory activity in NeuMitoFAP zebrafish exposed to MG2I and far-red light (Figures 4C-E), and (ii) ultrastructural changes in their mitochondria (Figure 5E, F).

As suggested, further evidence that the chemoptogenetic system targets mitochondria is now provided by new intravital imaging studies, showing mitochondrial fragmentation in lateral line nerve axons of live NeuMitoFAP-MG2I zebrafish immediately after far-red light exposure (new Figure 5A-D).

Together, these data (along with data in Figure 1 showing localization of the fluorogen activating protein dL5** within mitochondria) provide unambiguous evidence that the phenotypic changes we observed in FAP transgenic animals exposed to MG2I and far-red light are caused by singlet-oxygen induced deficits in mitochondrial structure and function.

3) The EM analyses were not well described. Such work – even with the extreme dysmorphology reported here – needs to be assessed in a blinded fashion. The authors could take the assembled scans, encode them, and have a blinded assessment to increase rigor of this work.

For each of 6 experimental groups (NeuMitoFAP-no chemical and NeuMitFAP-MG2I, prior to, immediately after, and 24 hours after light exposure), 12 identically-sized fields of view were re-imaged by electron microscopy. The resulting 72 images (containing a total of 69-83 mitochondria per group) were analyzed by an observer not involved in image acquisition and blinded to the identity of the sections (new Figure 5G). The number of mitochondria in each field was counted and each mitochondrion scored as ‘normal’ (multiple tubular or lamellar cristae, elongated shape) or ‘abnormal’ (absent or severely attenuated cristae, rounded shape). NeuMitoFAP-MG2I zebrafish exposed to far-red light showed 90.4 ± 3.5% (immediate post-light) and 95.0 ± 3.0% (24 hours post-light) abnormal mitochondria per section compared with 0 – 1.4% abnormal mitochondria in the control groups. These large differences were statistically significant, showing explicitly that the dramatic abnormalities observed in mitochondrial ultrastructure were specific to transgenic zebrafish exposed to MG2I and far-red light – and therefore attributable to chemoptogenetic damage.

4) Statistical analysis: The 1-way and/or 2-way ANOVA used to evaluate data in Figures 2D, 3B-C, 5A, 5C-D, and Figure 2—figure supplement 2B is not sufficient to assess differences between groups. A post hoc analysis with correction for multiple comparisons is required, as per Figure 2—figure supplement 2B, Figure 2—figure supplement 3B-C and Figure 4—figure supplement 1B. Figure legends should indicate if dots in scatter plots represent individual fish of a single exemplary experiment or across all replicates.

The appropriate post hoc tests were completed previously, but this information was unintentionally omitted from some of the figure legends. Missing details of post hoc tests and the source of data shown in scatterplots have now been added to the figure legends.

References:

1) He, J., et al. A genetically targetable near-infrared photosensitizer. Nat Methods 13, 263-268 (2016).

2) Qian, W., et al. Chemoptogenetic damage to mitochondria causes rapid telomere dysfunction. Proc Natl Acad Sci U S A 116, 18435-18444 (2019).